# Prompt Optimization Meets Subspace Representation Learning for Few-shot Out-of-Distribution Detection

## Abstract

The reliability of artificial intelligence (AI) systems in open-world settings depends heavily on their ability to flag out-of-distribution (OOD) inputs unseen during training. Recent advances in large-scale vision-language models (VLMs) have enabled promising few-shot OOD detection frameworks using only a handful of in-distribution (ID) samples. However, existing prompt learning-based OOD methods largely overlook the geometry of the visual feature embeddings learned by VLMs whose structure is particularly informative for distinguishing ID from OOD data and holds rich representation capacity as they are pre-trained on millions of samples. To address this, we introduce a *geometry-aware context optimization framework* that integrates subspace representation learning with prompt tuning. By projecting ID-relevant features into a subspace spanned by prompt vectors and simultaneously projecting ID-irrelevant components via orthogonal null-space projections, our approach strengthens the discriminative power of the learned prompt vectors, thereby leading to enhanced ID–OOD separability at test time. To enable an easy-to-handle, end-to-end learning under this framework, we design a geometry-regularized learning criterion that ensures strong OOD detection performance as well as high ID classification accuracy across settings. Moreover, the proposed framework can be seamlessly integrated with a wide range of existing context optimization methods, effectively complementing their softmax-based OOD detectors. Experiments on various real-world datasets showcase the effectiveness of our approach for reliable open-world AI systems.

## 1 Introduction

Deep learning models often exhibit overconfidence when exposed to inputs from unseen, out-of-distribution (OOD) categories (Goodfellow et al., 2014). Such overconfidence can lead to critical failures in open-world and safety-sensitive applications such as autonomous driving (Geiger et al., 2012) and medical diagnostics (Schlegl et al., 2017). These risks have spurred substantial interest in OOD detection approaches, that aim to equip models with the ability to reliably detect OOD inputs that falls outside the known class (Yang et al., 2024a). Traditional OOD detection approaches (Liu et al., 2020a; Huang et al., 2021; Lee et al., 2018; Huang & Li, 2021) typically rely on designing scoring functions or incorporating auxiliary outlier datasets during training. While such methods have demonstrated promise in controlled settings, they often fail to generalize in dynamic, real-world environments where the nature of the OOD data is unpredictable (Shen et al., 2024; Kirichenko et al., 2020; Fang et al., 2025).

Recently, large-scale vision-language models (VLMs) such as contrastive language-image pretraining (CLIP) (Radford et al., 2021) have shown strong zero-shot performance on downstream tasks by aligning visual and textual modalities in a shared embedding space. This opens a new direction for OOD detection, particularly in low-resource or few-shot settings (Esmaeilpour et al., 2022; Ming et al., 2022; Miyai et al., 2023b). However, CLIP's zero-shot approach depends heavily on manually crafted prompts, where even slight variations (e.g., "*a flower*" vs. "*a type of a flower*") can significantly impact performance (Yuksekgonul et al., 2022; Nie et al., 2024). To reduce this sensitivity, a class of prompt tuning methods called *context optimization* has been introduced. For example, CoOp (Zhou et al., 2022b) and CoCoOp (Zhou et al., 2022a) replace hand-crafted textual embed-

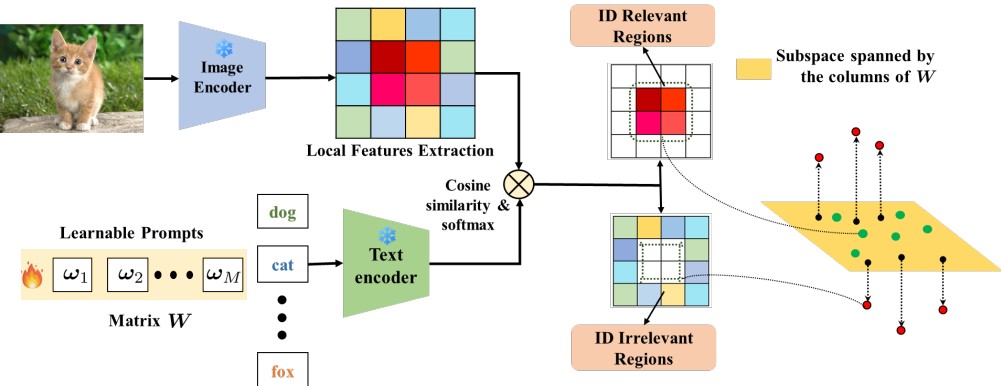

Figure 1: The proposed **Sub**space learning-based **Co**ntext **Op**timization (SubCoOp) framework for prompt-learning-based OOD detection.

dings with learnable context vectors that are optimized to enhance alignment between in-distribution (ID) image features and class text embeddings, leading to improved classification accuracy.

However, context optimization methods face a significant limitation in their direct applicability towards OOD detection tasks. By focusing on bringing ID image features closer to their text embeddings, these methods risk inadvertently incorporating background clutter or semantically irrelevant regions—some of which may actually represent OOD samples—into the ID representation space. This eventually weakens the model's ability to accurately distinguish between ID and OOD samples at test time. As a result, several subsequent approaches have introduced *proxy-OOD supervision* to explicitly guide models in learning more robust boundaries between ID and OOD samples (Miyai et al., 2023a; Yu et al., 2024; Xu et al., 2025). For instance, LoCoOp (Miyai et al., 2023a) addresses this limitation by leveraging CLIP's spatially-aware local features. It identifies ID-irrelevant regions—those where the true class is not among the top predictions—and treats them as proxy OOD features. By applying an entropy-maximization strategy to the predictions associated with ID-irrelevant features, this approach enhances the separation between ID and OOD samples without relying on any specific OOD data. A related method was proposed in (Yu et al., 2024), where adaptive weighting is incorporated into the LoCoOp optimization framework to dynamically balance ID- and OOD-specific loss terms based on the model's prediction confidence. Recently, the approach in (Xu et al., 2025) extends this idea incorporating pre-trained segmentation models for image inpainting to generate more informative proxy-OOD samples during few-shot training. Nevertheless, integrating this method into few-shot training is prohibitively expensive, requiring roughly 4–5 times longer training compared to other proxy-OOD supervision approaches (Miyai et al., 2023a; Yu et al., 2024), primarily due to the inpainting demands on the training dataset. Similarly, the approach in (Zeng et al., 2024) relies on extensive data augmentation and is more computationally expensive as it separately learns class-wise local prompts and introduces negative prompts as well. As a result, its overall training cost is approximately 8-9 times higher than these proxy-OOD supervision approaches (Miyai et al., 2023a; Yu et al., 2024).

**Our contributions.** In this work, we aim to enhance the OOD detection capabilities of the existing context optimization methods by efficiently extracting more informative proxy-OOD supervision through geometry-aware prompt tuning from the pretrained visual-textual CLIP encoder under a limited training sample budget. Existing prompt learning approaches, through their cosine similary-based cross-entropy loss training, primarily shape the relative geometry among ID classes, but nonetheless overlook the discriminative geometry between ID and OOD features. Towards this, we introduce a novel framework that explicitly leverages the inherently discriminative geometry of the visual feature embeddings of the ID and OOD features. As prompt vectors are the only learnable parameters in such frameworks, our key idea is to inject feature geometry-aware discriminative cues into their learning, thereby improving the ID-OOD separability at test time. Our key contributions are summarized as follows:

*(i).* To learn geometry-aware prompt vectors, we introduce subspace representation learning-based framework by projecting the ID features into a subspace spanned by the prompt vectors, while simultaneously projecting ID-irrelevant features into the orthogonal null space. This subspace-based formulation is designed to exploit a discriminative geometry between ID and OOD samples.

*(ii).* We design an easy-to-implement, subspace regularization loss that can be seamlessly integrated within context optimization, thereby enhancing the OOD detection performance without compromising the ID classification accuracy and without incurring any significant computational cost.

*(iii).* Experiments on large-scale real-world datasets such as ImageNet-1k (Deng et al., 2009) demonstrate that our method outperforms many state-of-the-art context optimization approaches for OOD detection and consistently performs well across diverse challenging settings.

## 2 PROBLEM STATEMENT

Consider an ID dataset $\mathcal{D}^{\text{in}} = \{(\boldsymbol{x}, y)\}$, where $\boldsymbol{x} \in \mathbb{R}^L$ denotes the input features of an image and $y \in \mathcal{Y}^{\text{in}} := \{1, \ldots, K\}$ is its corresponding class label (also referred to as the *true label*). AI models are typically trained under the closed-world assumption, where test samples are expected to come from the same distribution as the ID data. In practice, however, models frequently encounter OOD samples—data that deviates from the training distribution (Hendrycks & Gimpel, 2016). In classification settings, there may occur a semantic-shift such that test samples may belong to an *unknown* label space $\mathcal{Y}^{\text{out}}$, where $\mathcal{Y}^{\text{in}} \cap \mathcal{Y}^{\text{out}} = \emptyset$. The objective of OOD detection is to build a classifier that, given a test sample $\boldsymbol{x}$, predicts whether it belongs to an ID class or not, thereby preventing models from assigning high-confidence predictions to OOD samples. OOD detection can be framed as a binary classification problem. Formally, this is achieved through a detection function $\boldsymbol{d}_\eta : \mathbb{R}^L \to \{\text{ID}, \text{OOD}\}$ such that

$$\boldsymbol{d}_\eta(\boldsymbol{x}) = \begin{cases} \text{ID} & s(\boldsymbol{x}) \geq \eta \\ \text{OOD} & s(\boldsymbol{x}) < \eta, \end{cases} \tag{1}$$

where $s(\boldsymbol{x})$ is a scoring function associated with the input feature $\boldsymbol{x}$ and $\eta$ is the threshold.

**Context Optimization with Learnable Prompts.** Context optimization (CoOp) (Zhou et al., 2022b) leverages pre-trained VLMs, such as CLIP (Radford et al., 2021), for open-vocabulary visual recognition tasks. While CLIP typically uses static, hand-crafted prompts, CoOp learns a set of positive prompt vectors in a data-driven manner. These vectors are optimized as part of the model parameters during training, enabling few-shot learning for the downstream task.

Consider the ID input image $\boldsymbol{x} \in \mathbb{R}^L$, which is inputted to the visual encoder $\boldsymbol{f} : \mathbb{R}^L \to \mathbb{R}^D$ of CLIP to extract the visual feature vector $\boldsymbol{f}^{\text{in}} = \boldsymbol{f}(\boldsymbol{x})$. The textual prompt is composed as $\boldsymbol{t}_k = \{\boldsymbol{\omega}_1, \boldsymbol{\omega}_2, \ldots, \boldsymbol{\omega}_M, \boldsymbol{c}_k\}$, where each $\boldsymbol{\omega}_m \in \mathbb{R}^D$ is a learnable context vector, $\boldsymbol{c}_k \in \mathbb{R}^D$ is the class name embedding of the image, for each class $k \in [K]$, and $M$ is the number of prompt vectors. The textual encoder $\boldsymbol{g}$ processes the prompt $\boldsymbol{t}_k$ to yield the textual feature $\boldsymbol{g}_k = \boldsymbol{g}(\boldsymbol{t}_k)$ (e.g., using a Transformer-based model). With these notations, we can represent the class prediction probabilities $\Pr(y = k \mid \boldsymbol{x})$ as follows:

$$\Pr(y = k \mid \boldsymbol{x}) = \frac{\exp\left(\text{sim}(\boldsymbol{f}^{\text{in}}, \boldsymbol{g}_k)/\tau\right)}{\sum_{k'=1}^{K} \exp\left(\text{sim}(\boldsymbol{f}^{\text{in}}, \boldsymbol{g}_{k'})/\tau\right)}, \tag{2}$$

where $\text{sim}(\cdot, \cdot)$ denotes the cosine similarity and $\tau > 0$ is a temperature parameter. Consequently, CoOp optimizes the prompt vectors using the cross-entropy loss by matching the class predictions in equation 2 and the true label $y$, i.e., $\mathcal{L}_{\text{CE}} = -\sum_{k=1}^{K} \mathbb{I}[y = k] \log \Pr(y = k \mid \boldsymbol{x})$. Although CoOp aligns the ID image with its class text embedding $\boldsymbol{g}_k$ in this manner, it inadvertently brings the text embedding closer to background or ID-irrelevant features with the ID image, resulting in incorrectly high confidence scores for OOD images during test time. Hence, without access to OOD samples during training, the model struggles to learn a well-defined ID-OOD boundary for reliable OOD detection.

**OOD Local Features Extraction.** Recently, LoCoOp (Miyai et al., 2023a) introduced a novel perspective to prompt optimization-based OOD detection by extracting local features that serve as proxy OOD signals, thereby preventing the model from assigning high ID confidence scores to OOD-like features. To detect local features not corresponding to ID classes (i.e., ID-irrelevant features), the method in (Miyai et al., 2023a) examine a set of spatial indices from the feature map: $\mathcal{I} = \{0, 1, 2, \ldots, H \times W - 1\}$, where $H$ and $W$ are the height and width of the feature map, respectively. Following a strategy inspired by semantic segmentation (Radford et al., 2021), the

class probabilities associated to each region $i \in \mathcal{I}$ can be computed based on the similarity between local visual features and text embeddings:

$$\mathsf{Pr}_i(y = k \mid \boldsymbol{x}) = \frac{\exp\left(\mathrm{sim}(\boldsymbol{f}_i^{\mathrm{in}}, \boldsymbol{g}_k)/\tau\right)}{\sum_{k'=1}^{K} \exp\left(\mathrm{sim}(\boldsymbol{f}_i^{\mathrm{in}}, \boldsymbol{g}_{k'})/\tau\right)}, \tag{3}$$

where $\boldsymbol{f}_i^{\mathrm{in}} \in \mathbb{R}^D$ denotes the feature extracted from the $i$th local region of the image $\boldsymbol{x}$ and $\boldsymbol{g}_k$ corresponds to the text prompt embedding for the $k$th class as defined in equation 2.

For any region $i$ of the image $\boldsymbol{x}$, if it corresponds to the ID class, its ground-truth label $y$ is expected to appear among the top-$C$ predicted classes. Conversely, if the region is unrelated to any ID class (e.g., background noise), the true class is unlikely to rank within the top-$C$, due to the lack of strong semantic alignment. Leveraging this observation, one can define an index set $\mathcal{J}$ to identify such ID-irrelevant regions:

$$\mathcal{J} = \{i \in \mathcal{I} : \mathrm{rank}\left(\mathsf{Pr}_i(y \mid \boldsymbol{x})\right) > C\}. \tag{4}$$

Here, $\mathrm{rank}\left(\mathsf{Pr}_i(y \mid \boldsymbol{x})\right)$ denotes the rank of the true label $y$ among the predicted scores over all ID classes and $C$ is a hyperparameter or can be fixed based on prior knowledge about the number of fine-grained classes or semantic relationships in the dataset. The methods in (Miyai et al., 2023a; Yu et al., 2024; Xu et al., 2025) utilized such extracted ID-irrelevant features to increase the uncertainty of their softmax-based class probability predictions using an entropy regularization (ER) given by:

$$\mathcal{L}_{\mathrm{Ent}} = -\sum_{i \in \mathcal{J}} H\left(\boldsymbol{p}_i(\boldsymbol{x})\right), \tag{5}$$

where $H(\boldsymbol{p}) = -\sum_{k=1}^{K} p_k \log p_k$ denotes entropy function and $\boldsymbol{p}_i(\boldsymbol{x})$, $i \in \mathcal{J}$ is a $K$-dimensional probability vector, where each entry represents $\mathsf{Pr}_i(y = k \mid \boldsymbol{x})$, as defined in equation 3.

## 3 PROPOSED APPROACH

**Motivation.** The idea of using proxy OOD signals derived from ID-irrelevant regions in the training data is promising—especially since OOD data is typically unavailable at test time. Nonetheless, extracting more informative proxy OOD supervision is crucial as the budget of the training samples is limited under few-shot settings. Existing methods (Miyai et al., 2023a; Yu et al., 2024; Xu et al., 2025) rely on class prediction probabilities from ID/OOD regions to train the model to distinguish between ID and OOD data at test time. Yet, the high dimensional feature embeddings of the training data is much more informative—that is largely overlooked in the current approaches. In this context, a more robust and generalizable alternative would be to further incorporate unsupervised characterization techniques that captures the geometry of the feature representations. This could make extraction of proxy-OOD supervision more effective in few-shot settings without incurring much computational overhead.

**Our Idea: Prompt Vectors-induced Subspace Projection.** To enhance the OOD detection on the prompt learning-based approaches, we propose to leverage subspace projection techniques on the extracted local regions of the training data. Prior work indicates that pretrained VLM embeddings (e.g., CLIP) for ID data exhibits *low-dimensionality* due to their class-informative nature (Zhu et al., 2023; Bhalla et al., 2024). We aim to exploit this geometry by learning a low-dimensional basis $\boldsymbol{W} \in \mathbb{R}^{D \times M}$ that spans the ID subspace. During optimization, we aim to increase the alignment of ID regions to the subspace spanned by $\boldsymbol{W}$ and simultaneously inflate its orthogonal residual components for OOD regions. To this end, we parameterize the basis $\boldsymbol{W}$ with the same prompt vectors $\boldsymbol{\omega}_1, \ldots, \boldsymbol{\omega}_M$ used for context optimization (see equation 2), yielding a parameter-efficient design. In this way, the learned prompt vectors preserve the ID–OOD separating geometry of the feature space alongside the class-informative geometry induced by cosine similarity matching as defined in equation 2.

Consider the matrix formed by the prompt vectors $\boldsymbol{\omega}_1, \ldots, \boldsymbol{\omega}_M$, i.e., $\boldsymbol{W} = [\boldsymbol{\omega}_1, \boldsymbol{\omega}_2, \ldots, \boldsymbol{\omega}_M]$. We project the local feature vectors corresponding to ID data onto an $M$-dimensional subspace spanned by the column vectors of $\boldsymbol{W} \in \mathbb{R}^{D \times M}$, also called the ID subspace and denoted as $\mathcal{R}(\boldsymbol{W})$. At the same time, the features from ID-irrelevant or OOD regions are projected to lie in the null space $\mathcal{N}(\boldsymbol{W})$ orthogonal to $\mathcal{R}(\boldsymbol{W})$, defined as $\mathcal{N}(\boldsymbol{W}) = \left\{\boldsymbol{f} \in \mathbb{R}^D : \boldsymbol{W}^\top \boldsymbol{f} = \boldsymbol{0}\right\}$, which has dimension

Table 1: OOD detection performance of our method and the baselines on various OOD datasets. Here ID dataset is ImageNet-1k. All methods employ the same CLIP-ViT-B/16 backbone. Results with ⋆ marked are taken from (Miyai et al., 2023a; Yu et al., 2024).

| Method | iNaturalist | | SUN | | Places365 | | Textures | | Average | |
|---|---|---|---|---|---|---|---|---|---|---|
| | FPR95↓ | AUROC↑ | FPR95↓ | AUROC↑ | FPR95↓ | AUROC↑ | FPR95↓ | AUROC↑ | FPR95↓ | AUROC↑ |
| *Zero-shot methods* | | | | | | | | | | |
| MCM⋆ | 30.94 | 94.61 | 37.67 | 92.56 | 44.76 | 89.76 | 57.91 | 86.10 | 42.82 | 90.76 |
| GL-MCM⋆ | 15.18 | 96.71 | 30.42 | 93.09 | 38.85 | 89.90 | 57.93 | 83.63 | 35.47 | 90.83 |
| *Post-hoc methods with fine-tuned CLIP* | | | | | | | | | | |
| MSP⋆ | 74.57 | 77.74 | 76.95 | 73.97 | 79.72 | 72.18 | 73.66 | 74.84 | 74.98 | 76.22 |
| ODIN⋆ | 98.93 | 57.73 | 88.72 | 78.42 | 87.80 | 76.88 | 85.47 | 71.49 | 90.23 | 71.13 |
| EnergyScore⋆ | 64.98 | 87.18 | 46.42 | 91.17 | 57.40 | 87.33 | 50.39 | 88.22 | 54.80 | 88.48 |
| ReAct⋆ | 65.57 | 86.87 | 46.17 | 91.04 | 56.85 | 87.42 | 49.88 | 88.13 | 54.62 | 88.37 |
| MaxLogit⋆ | 60.88 | 88.03 | 44.83 | 91.16 | 55.54 | 87.45 | 48.72 | 88.63 | 52.49 | 88.82 |
| *Prompt tuning-based methods (16-shot)* | | | | | | | | | | |
| LSN | $46.40^{\pm 1.76}$ | $91.91^{\pm 2.73}$ | $31.86^{\pm 1.56}$ | $93.21^{\pm 1.32}$ | $40.61^{\pm 0.65}$ | $90.05^{\pm 1.53}$ | $47.21^{\pm 0.88}$ | $88.98^{\pm 0.97}$ | $41.52^{\pm 1.21}$ | $91.04^{\pm 1.64}$ |
| NegPrompt | $38.11^{\pm 1.15}$ | $90.22^{\pm 0.78}$ | $31.44^{\pm 0.29}$ | $92.59^{\pm 0.18}$ | $36.15^{\pm 2.05}$ | $90.97^{\pm 0.78}$ | $44.64^{\pm 1.34}$ | $87.49^{\pm 0.52}$ | $37.59^{\pm 1.21}$ | $90.32^{\pm 0.57}$ |
| IDLike | $9.71^{\pm 0.60}$ | $98.05^{\pm 0.07}$ | $38.93^{\pm 0.10}$ | $90.54^{\pm 0.08}$ | $47.06^{\pm 1.44}$ | $88.06^{\pm 1.93}$ | $32.82^{\pm 5.12}$ | $91.89^{\pm 1.49}$ | $32.12^{\pm 1.09}$ | $92.14^{\pm 0.01}$ |
| CoOp | $26.72^{\pm 2.09}$ | $94.53^{\pm 0.36}$ | $36.96^{\pm 0.87}$ | $92.34^{\pm 0.15}$ | $45.01^{\pm 1.45}$ | $89.43^{\pm 0.15}$ | $40.38^{\pm 1.45}$ | $90.95^{\pm 0.18}$ | $37.27^{\pm 1.47}$ | $91.81^{\pm 0.21}$ |
| LoCoOp | $18.70^{\pm 2.12}$ | $96.09^{\pm 0.38}$ | $22.83^{\pm 0.98}$ | $95.12^{\pm 0.07}$ | $34.78^{\pm 3.47}$ | $91.52^{\pm 0.63}$ | $43.75^{\pm 0.22}$ | $89.81^{\pm 0.33}$ | $30.02^{\pm 1.70}$ | $93.14^{\pm 0.35}$ |
| SCT | $16.14^{\pm 1.81}$ | $96.68^{\pm 0.29}$ | $21.57^{\pm 1.20}$ | $95.23^{\pm 0.26}$ | $31.47^{\pm 0.89}$ | $91.89^{\pm 0.25}$ | $43.75^{\pm 0.56}$ | $88.83^{\pm 0.45}$ | $28.23^{\pm 1.12}$ | $93.16^{\pm 0.31}$ |
| SubCoOp | $12.61^{\pm 1.69}$ | $97.28^{\pm 0.38}$ | $18.75^{\pm 1.47}$ | $95.82^{\pm 0.20}$ | $29.45^{\pm 1.66}$ | $92.51^{\pm 0.13}$ | $41.06^{\pm 1.02}$ | $90.65^{\pm 0.25}$ | $\mathbf{25.47^{\pm 1.46}}$ | $\mathbf{94.07^{\pm 0.24}}$ |

Table 2: OOD detection performance of various prompt tuning-based approaches with and without subspace regularizations in 16-shot settings. Here ID dataset is ImageNet-1k dataset.

| Method | iNaturalist | | SUN | | Places365 | | Texture | | Average | |
|---|---|---|---|---|---|---|---|---|---|---|
| | FPR95↓ | AUROC↑ | FPR95↓ | AUROC↑ | FPR95↓ | AUROC↑ | FPR95↓ | AUROC↑ | FPR95↓ | AUROC↑ |
| CoOp | $14.70^{\pm 2.28}$ | $96.40^{\pm 0.65}$ | $28.07^{\pm 1.65}$ | $92.60^{\pm 0.72}$ | $37.37^{\pm 2.01}$ | $89.78^{\pm 0.68}$ | $43.55^{\pm 1.95}$ | $87.55^{\pm 0.72}$ | $30.92^{\pm 1.97}$ | $91.58^{\pm 0.69}$ |
| CoOp-SR | $14.85^{\pm 3.36}$ | $96.76^{\pm 0.94}$ | $25.19^{\pm 0.77}$ | $94.62^{\pm 0.07}$ | $33.34^{\pm 0.91}$ | $91.24^{\pm 0.19}$ | $41.85^{\pm 1.22}$ | $89.74^{\pm 0.49}$ | $\mathbf{28.81^{\pm 1.57}}$ | $\mathbf{93.59^{\pm 0.42}}$ |
| LoCoOp | $18.70^{\pm 2.12}$ | $96.09^{\pm 0.38}$ | $22.83^{\pm 0.98}$ | $95.12^{\pm 0.07}$ | $34.78^{\pm 3.47}$ | $91.52^{\pm 0.63}$ | $43.75^{\pm 0.22}$ | $89.81^{\pm 0.33}$ | $30.02^{\pm 1.70}$ | $93.14^{\pm 0.35}$ |
| LoCoOp-SR | $14.33^{\pm 0.76}$ | $96.99^{\pm 0.08}$ | $22.14^{\pm 1.96}$ | $95.10^{\pm 0.44}$ | $32.04^{\pm 2.82}$ | $92.07^{\pm 0.61}$ | $42.35^{\pm 3.04}$ | $89.87^{\pm 0.53}$ | $\mathbf{27.72^{\pm 2.15}}$ | $\mathbf{93.51^{\pm 0.42}}$ |
| SCT | $16.14^{\pm 1.81}$ | $96.68^{\pm 0.29}$ | $21.57^{\pm 1.20}$ | $95.23^{\pm 0.26}$ | $31.47^{\pm 0.89}$ | $91.89^{\pm 0.25}$ | $43.75^{\pm 0.56}$ | $88.83^{\pm 0.45}$ | $28.23^{\pm 1.12}$ | $93.16^{\pm 0.31}$ |
| SCT-SR | $12.61^{\pm 1.69}$ | $97.28^{\pm 0.38}$ | $18.75^{\pm 1.47}$ | $95.82^{\pm 0.20}$ | $29.45^{\pm 1.66}$ | $92.51^{\pm 0.13}$ | $41.06^{\pm 1.02}$ | $90.65^{\pm 0.25}$ | $\mathbf{25.47^{\pm 1.46}}$ | $\mathbf{94.07^{\pm 0.24}}$ |
| OSPCoOp | $14.28^{\pm 1.37}$ | $97.11^{\pm 0.35}$ | $18.95^{\pm 1.25}$ | $96.52^{\pm 0.07}$ | $27.18^{\pm 1.37}$ | $93.52^{\pm 0.57}$ | $41.75^{\pm 0.25}$ | $90.96^{\pm 0.31}$ | $25.54^{\pm 1.06}$ | $94.53^{\pm 0.33}$ |
| OSPCoOp-SR | $12.89^{\pm 1.73}$ | $97..41^{\pm 0.41}$ | $18.02^{\pm 1.36}$ | $96.69^{\pm 0.12}$ | $26.94^{\pm 1.22}$ | $93.46^{\pm 0.52}$ | $40.79^{\pm 0.22}$ | $90.93^{\pm 0.35}$ | $\mathbf{24.66^{\pm 1.13}}$ | $\mathbf{94.62^{\pm 0.35}}$ |

$D - M$. It is important to keep $M < D$, since when $M = D$, the null space becomes trivial (containing only the zero vector), thus limiting our ability to separate ID and OOD features effectively. This condition is typically satisfied in practice, as the number of prompt vectors $M$ is usually small (e.g., $M \approx 16$ as suggested in (Zhou et al., 2022b; Miyai et al., 2023a)), whereas the dimensionality of CLIP embeddings is relatively large (e.g., $D = 512$). Based on these complementary projections, we propose *subspace regularizations* (SR) for the ID and OOD regions as follows:

$$\mathcal{L}_{\text{Sub-ID}} = \sum_{i \in \mathcal{J}'} \frac{\left\| \text{Proj}_{\boldsymbol{W}^{\perp}} \left( \boldsymbol{f}_i^{\text{in}} \right) \right\|_2}{\left\| \boldsymbol{f}_i^{\text{in}} \right\|_2}, \ \mathcal{L}_{\text{Sub-OOD}} = \sum_{i \in \mathcal{J}} \frac{\left\| \text{Proj}_{\boldsymbol{W}} \left( \boldsymbol{f}_i^{\text{in}} \right) \right\|_2}{\left\| \boldsymbol{f}_i^{\text{in}} \right\|_2}, \quad (6)$$

where $\boldsymbol{f}_i^{\text{in}}$ denotes the $i$th local region feature for the data item $\boldsymbol{x}$, $\mathcal{J}'$ is the complement of the set $\mathcal{J}$, i.e., $\mathcal{J}' = \mathcal{I} \setminus \mathcal{J} = \{i \in \mathcal{I} \mid i \notin \mathcal{J}\}$, and the projections $\text{Proj}_{\boldsymbol{W}^{\perp}}$ and $\text{Proj}_{\boldsymbol{W}}$ are given by:

$$\text{Proj}_{\boldsymbol{W}^{\perp}}(\boldsymbol{f}) = \left( \boldsymbol{I}_D - \boldsymbol{W} \left( \boldsymbol{W}^{\top} \boldsymbol{W} \right)^{-1} \boldsymbol{W}^{\top} \right) \boldsymbol{f}, \ \text{Proj}_{\boldsymbol{W}}(\boldsymbol{f}) = \left( \boldsymbol{W} \left( \boldsymbol{W}^{\top} \boldsymbol{W} \right)^{-1} \boldsymbol{W}^{\top} \right) \boldsymbol{f}. \quad (7)$$

Here, the loss term $\mathcal{L}_{\text{Sub-ID}}$ encourages ID features to lie within the column space $\mathcal{R}(\boldsymbol{W})$ by minimizing their projected components in the orthogonal complement, $\mathcal{N}(\boldsymbol{W})$. Conversely, the loss term $\mathcal{L}_{\text{Sub-OOD}}$ promotes OOD features to lie in $\mathcal{N}(\boldsymbol{W})$ by suppressing their projections onto $\mathcal{R}(\boldsymbol{W})$.

**Implementation.** The proposed SRs in equation 6 can be easily integrated to the context optimization framework. As a result, we propose the following *geometry-aware prompt learning* criterion that combines the cross-entropy loss with the SRs as follows:

$$\mathcal{L} = (1 - \text{Pr}(y|\boldsymbol{x})) \cdot \mathcal{L}_{\text{CE}} + \text{Pr}(y|\boldsymbol{x}) \cdot (\lambda_1 \mathcal{L}_{\text{Sub-ID}} + \lambda_2 \mathcal{L}_{\text{Sub-OOD}} + \lambda_3 \mathcal{L}_{\text{Ent}}) \quad (8)$$

where $(\boldsymbol{x}, y)$ denotes the image-label pair of the ID data, $\mathcal{L}_{\text{CE}}$ is the cross-entropy loss as defined after equation 2, other regularization terms are defined in equation 6 and equation 5, and $\lambda_1, \lambda_2, \lambda_3 > 0$ are the respective regularization parameters. Here, we employ the modulation weights using the

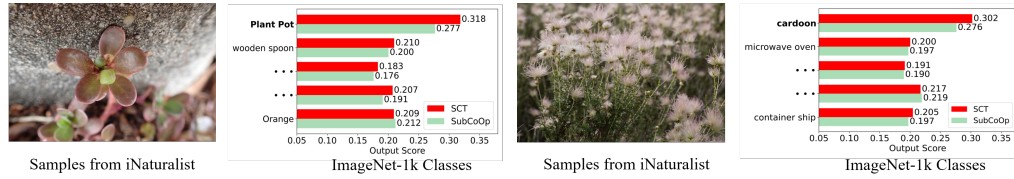

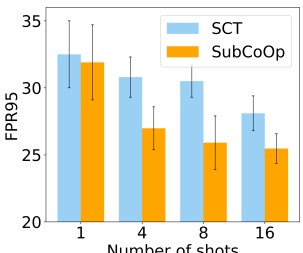 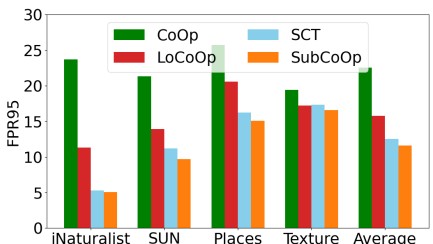

Figure 2: Example images from the iNaturalist dataset that are visually and semantically similar to certain ImageNet-1k classes. Comparison of similarity scores from SCT and our proposed Sub-CoOp. While SCT assigns high similarity scores to the ImageNet-1k ID classes, leading to incorrect detection as ID, SubCoOp effectively suppresses such scores, enabling the correct OOD detection.

Figure 3: OOD detection performance of various few-shot techniques in ImageNet-1k dataset

Figure 4: OOD performance of our method SubCoOp and other methods across various OOD datasets with ID dataset as ImageNet-100

softmax probabilities $\Pr(y|\boldsymbol{x})$, as proposed in the recent work (Yu et al., 2024). This reweighting strategy enables dynamic adjustment of the classification and regularization loss contributions according to the model's confidence in its predictions. This implies that, when the model exhibits lower confidence, the contributions from both SR and ER losses are downweighted, as ID-irrelevant region selection in equation 4 becomes less reliable.

The final training loss averages $\mathcal{L}$ across all the training examples and can be easily learned using backpropagation-based optimizers in an end-to-end manner. We refer our approach (also see Fig. 1) as **Sub**space learning-based **Co**ntext **Op**timization (SubCoOp).

**Remark 1** *As one can see, the SR computations do not introduce significant computational overhead, as the per-feature cost for the projection operation is $\mathcal{O}(dM + M^2)$, which is typically dominated by $\mathcal{O}(dM)$ since $M$ is much smaller than the CLIP embedding dimension $D$. Consequently, this cost is negligible compared to the CLIP forward pass required to produce the feature embeddings. Moreover, to maintain $\boldsymbol{W}$ as a full-rank matrix with independent vectors representing the basis of the ID subspace, we could impose explicit orthogonality or full-rank regularizations (e.g., nuclear norm or log-determinant norm). However, in practice, a soft constraint using $\ell_2$ regularization (weight decay) often suffices, as it encourages more uniform singular values across the $M$ dimensions without introducing additional complex regularization terms, as shown in our experiments. Also, to ensure the matrix inversions in equation 7 is well-conditioned, we compute $(\boldsymbol{W}^\top \boldsymbol{W} + \epsilon I_M)^{-1}$, where $\epsilon > 0$ is a small scalar that helps prevent rank deficiency and improve numerical stability.*

## 4 EXPERIMENTS

### 4.1 SETUP

**Dataset.** We employ ImageNet-1k and ImageNet-100 datasets (Deng et al., 2009) as ID data. For OOD data, we use a number of commonly used benchmark datasets such as iNaturalist (Van Horn, 2018), SUN (Xiao et al., 2010), Places Zhou et al. (2017), and Texture (Van Horn et al., 2018). For the few-shot training, we use 1-16 images per ID class, and evaluate the model using the whole OOD datasets and the test ID dataset.

**Implementation Details.** We employ the ViT-B/16 model (Dosovitskiy et al., 2020) as the backbone of the visual encoder for the pretrained CLIP model. For ID-irrelevant feature extraction, we set the

Table 3: OOD detection performance comparison with LoCoOp on hard OOD detection tasks. Bold numbers represent superior results.

| ID Dataset | OOD Dataset | Method | FPR95↓ | AUROC↑ |
|---|---|---|---|---|
| ImageNet-100 | ImageNet-10 | SCT | 46.05 | 88.37 |
| | | SubCoOp | **44.34** | **88.58** |
| ImageNet-20 | ImageNet-10 | SCT | 10.02 | **97.96** |
| | | SubCoOp | **9.15** | 97.72 |
| ImageNet-10 | ImageNet-20 | SCT | 14.71 | 95.64 |
| | | SubCoOp | **12.63** | **95.92** |
| ImageNet-10 | ImageNet-100 | SCT | 6.42 | 97.75 |
| | | SubCoOp | **5.92** | **97.95** |
| ImageNet-100 | ImageNet-20 | SCT | 58.53 | 81.19 |
| | | SubCoOp | **57.17** | **81.34** |

Table 4: OOD detection performance of SubCoOp under different SR and ER settings. ID dataset is ImageNet-1k. Here, × indicates a zero regularization parameter, and ✓ indicates a non-zero value.

| $\lambda_1$ $\lambda_2$ $\lambda_3$ | iNaturalist | | SUN | | Places | | Texture | | Average | |
|---|---|---|---|---|---|---|---|---|---|---|
| | FPR95↓ | AUROC↑ | FPR95↓ | AUROC↑ | FPR95↓ | AUROC↑ | FPR95↓ | AUROC↑ | FPR95↓ | AUROC↑ |
| × × × | $14.70^{\pm2.28}$ | $96.40^{\pm0.65}$ | $28.07^{\pm1.65}$ | $92.60^{\pm0.72}$ | $37.37^{\pm2.01}$ | $89.78^{\pm0.68}$ | $43.55^{\pm1.95}$ | $87.55^{\pm0.72}$ | $30.92^{\pm1.97}$ | $91.58^{\pm0.69}$ |
| × × ✓ | $16.14^{\pm1.81}$ | $96.68^{\pm0.29}$ | $21.57^{\pm1.20}$ | $95.23^{\pm0.26}$ | $31.47^{\pm0.89}$ | $91.89^{\pm0.25}$ | $43.75^{\pm0.56}$ | $88.83^{\pm0.45}$ | $28.23^{\pm1.12}$ | $93.16^{\pm0.31}$ |
| ✓ × ✓ | $14.12^{\pm1.29}$ | $97.61^{\pm0.17}$ | $20.62^{\pm1.40}$ | $95.77^{\pm0.29}$ | $30.16^{\pm0.83}$ | $92.42^{\pm0.22}$ | $42.64^{\pm0.27}$ | $89.15^{\pm0.21}$ | $26.89^{\pm0.95}$ | $93.74^{\pm0.22}$ |
| × ✓ ✓ | $15.45^{\pm2.43}$ | $96.89^{\pm0.55}$ | $20.52^{\pm1.82}$ | $95.61^{\pm0.32}$ | $30.12^{\pm1.58}$ | $92.48^{\pm0.26}$ | $43.21^{\pm0.43}$ | $88.82^{\pm0.51}$ | $27.33^{\pm1.57}$ | $93.45^{\pm0.41}$ |
| ✓ ✓ ✓ | $12.61^{\pm1.69}$ | $97.28^{\pm0.38}$ | $18.75^{\pm1.47}$ | $95.82^{\pm0.20}$ | $29.45^{\pm1.66}$ | $92.51^{\pm0.13}$ | $41.06^{\pm1.02}$ | $90.65^{\pm0.25}$ | $\mathbf{25.47^{\pm1.46}}$ | $\mathbf{94.07^{\pm0.24}}$ |

rank threshold parameter $C$ to the recommended value 100 and 20 for ImageNet-1k and ImageNet-100, respectively, based on the number of fine-grained classes (Miyai et al., 2023a). In addition, we fix $M = 16$, $\lambda_1 = 0.25$, $\lambda_2 = 2$, and $\lambda_3 = 5$, unless specified otherwise. We employ the SGD optimizer with a learning rate of 0.002, a batch size of 32, and train the model for 25 epochs. We use Nvidia 3090 Ti GPU for all the experiments.

**OOD Detection Score:** While testing, we adopt the global-local maximum concept matching (GL-MCM) score (Miyai et al., 2023b; 2025) for OOD detection (i.e., the score function $s(\boldsymbol{x})$ as employed in equation 1). This metric integrates the maximum softmax probability scores from both whole image feature and local image features and is defined as follows:

$$s_{\text{GL-MCM}}(\boldsymbol{x}) = \max_k \frac{\exp\left(\text{sim}(\boldsymbol{f}, \boldsymbol{g}_k)/\tau\right)}{\sum_{k'=1}^{K} \exp\left(\text{sim}(\boldsymbol{f}, \boldsymbol{g}_{k'})/\tau\right)} + \max_{k,i} \frac{\exp\left(\text{sim}(\boldsymbol{f}_i, \boldsymbol{g}_k)/\tau\right)}{\sum_{k'=1}^{K} \exp\left(\text{sim}(\boldsymbol{f}_i, \boldsymbol{g}_{k'})/\tau\right)} \tag{9}$$

where $\boldsymbol{f}$ is the vision encoder output for the test image $\boldsymbol{x}$, $\boldsymbol{f}_i$'s are its features corresponding to the local regions, and $\tau > 0$ denotes the temperature scaling parameter.

**Evaluation Metrics.** We evaluate the OOD detection performance using the following metrics: *(i)* FPR95 refers to the false positive rate (FPR) of OOD samples when the true positive rate (TPR) of ID samples is at 95%; *(ii)* area under the receiver operating characteristic curve (AUROC), measures the model's ability to distinguish between ID and OOD samples by evaluating TPR vs. FPR95 across all thresholds; and *(iii)* classification accuracy on ID data.

**Baselines.** To evaluate our proposed method, we consider a number of recently proposed prompt tuning-based approaches. Specifically, we employ LSN (Nie et al., 2024), NegPrompt (Liang et al., 2017), IDLike (Bai et al., 2024), CoOp (Zhou et al., 2022b), LoCoOp (Miyai et al., 2023a), and SCT (Yu et al., 2024). CoOp, LoCoOp, SCT and our approach SubCoOp are based on learning a set of positive prompts. On the other hand, NegPrompt and LSN each learn a set of negative prompts per ID class in addition to the positive prompt vectors. IDLike (Bai et al., 2024) is based on extracting outlier from ID data by performing spatial cropping on the images to enhance the OOD detection. In addition, we also compare with zero-shot approaches and post-hoc methods with CLIP fine tuning. For the zero shot baselines, we use the state-of-the-art MCM (Ming et al., 2022) and GL-MCM (Miyai et al., 2023b) methods. For the post-hoc methods, we adopt a number of popular OOD scoring methods such as MSP (Hendrycks & Gimpel, 2016), ReAcT (Sun et al., 2021), ODIN (Liang et al., 2017), MaxLogit Basart et al. (2022), and Energy Score (Liu et al., 2020b). These methods leverage CLIP's fine-tuned representations and combine them with simple post-processing techniques/scores for OOD detection. Unless otherwise specified, we adopt the same OOD detec-

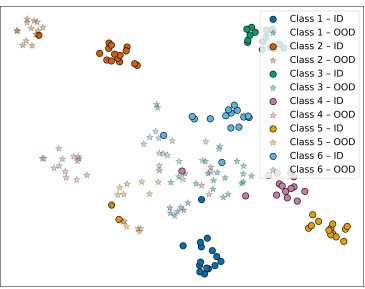 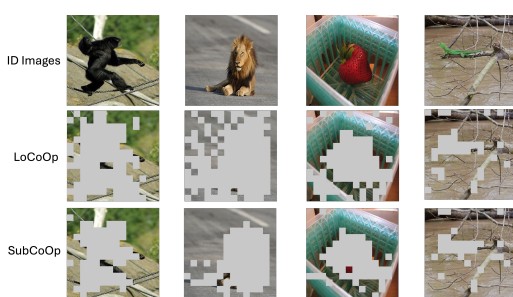

Figure 5: UMAP visualization of local ID and OOD local feature embeddings extracted by SubCoOp where we randomly choose 6 classes from the ImageNet-1K dataset for easy visualization.

Figure 6: Visualization of extracted local ID (gray patches) and OOD regions (colored regions) with LoCoOp and SubCoOp method.

tion score functions originally proposed by the respective methods. For instance, CoOp utilizes the MCM score, while both LoCoOp and SCT employ the GL-MCM score, similar to our approach SubCoOp. In addition, we include the recently proposed baseline OSPCoOp (Xu et al., 2025) in selected experiments, as its in-built segmentation-based inpainting makes it computationally expensive.

## 4.2 MAIN RESULTS

In Table 1, we present the OOD detection performance of our proposed approach and the baselines with ImageNet-1k as the ID dataset under different OOD datasets. The results are averaged over three random trials and the standard deviation is also reported. One can note that prompt tuning–based methods outperform other line of approaches as they encourage the model to align visual features with more discriminative and dynamically learned text prompts. More importantly, our proposed method SubCoOp outperforms the state-of-the-art prompt learning-based approaches with a notable margin. SubCoOp attains the best OOD detection performance, with a reduction of 2.76% in FPR95 and an improvement of 0.92% in AUROC compared to the next best performing method SCT. Our method particularly exhibit substantial improvements on challenging datasets such as iNaturalist and Places365, with an average FPR95 reduction of 3.53% and 2.82%, respectively compared to the SCT method. To provide a qualitative comparison, we present the class prediction probabilities output by SubCoOp, and by SCT on a few OOD samples from iNaturalist that is semantically similar to certain ID classes from ImageNet-1k, as shown in Fig. 2. SubCoOp also maintains high ID classification accuracy as shown in the supplementary material.

We further analyze the advantages of our proposed subspace regularizations (SR) by incorporating into other prompt-learning approaches. For example, the method CoOp (Zhou et al., 2022b) with SR is trained with the following modified objective function: $\mathcal{L}_{\text{CoOp-SR}} = \mathcal{L}_{\text{CE}} + \lambda_1 \mathcal{L}_{\text{Sub-ID}} + \lambda_2 \mathcal{L}_{\text{Sub-OOD}}$, where $\mathcal{L}_{\text{Sub-ID}}$ and $\mathcal{L}_{\text{Sub-OOD}}$ are defined in equation 6. Similarly, we can easily incorporate the proposed SR into other prompt learning-based approaches. In Table 2, we present the results that analyze the performance enhancement by the proposed subspace regularizations on various prompt learning techniques. For a fair comparison and to specifically highlight the contribution of the proposed SR, we employ the GL-MCM score for all the methods in Table 2. One can note that in all the cases, the proposed regularization improved the OOD performance by noticeable margin. For example, in the case of CoOp, CoOp-SR reduces the average FPR95 from 30.92% to 28.81% and boosts the average AUROC from 91.58% to 93.59%, with significant gains on the SUN and Places365 datasets. Similar performance gains are observed the case of LoCoOp method as well, further reinforcing the consistent performance enhancement by SR.

In Fig. 3, we compare the OOD detection performance of our method SubCoOp and the competing baseline SCT under varying few-shot settings with ImageNet-1k as the ID data set. Specifically, we present the average FPR95 and AUROC scores across all the OOD datasets under test. Both methods demonstrate consistent improvements in detection performance as the number of shots increases. Sub-CoOp generally outperforms SCT, particularly in the higher-shot settings, showing lower average FPR95 and higher average AUROC. Fig. 4 presents the OOD detection performance

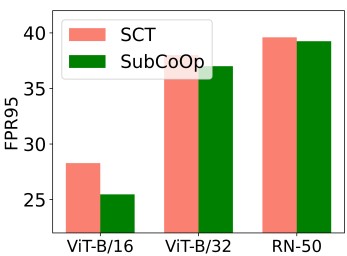

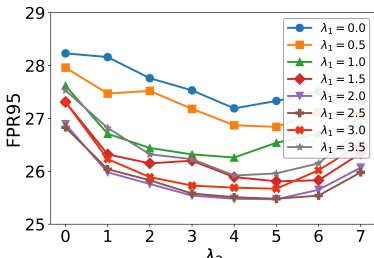

Figure 7: Average OOD detection performance across different image encoders for ImageNet-1k dataset

Figure 8: Average OOD detection performance of SubCoOp across different values of $\lambda_1$ and $\lambda_2$ using ImageNet-1k as the ID dataset.

of various prompt learning methods under the 16-shot setting, using ImageNet-100 as the ID dataset. SubCoOp outperforms all baselines under test, achieving the lowest average FPR95 (11.60%) and the highest average AUROC (97.80%). Compared to SCT and LoCoOp, SubCoOp demonstrates consistent detection performance across all OOD datasets. More results and discussion related to ImageNet-100 are relegated to supplementary section.

Figure 5 visualizes the UMAP projection of local feature embeddings extracted by SubCoOp from both ID-relevant and ID-irrelevant (proxy OOD) regions. We observe that ID-relevant features form compact and well-separated class clusters, whereas the ID-irrelevant features are distributed farther from their corresponding class centers and exhibit reduced overlap with ID clusters. This highlights SubCoOp's effectiveness in promoting a discriminative subspace for ID/OOD separation. This geometry, which was not exploited in prior approaches like LoCoOp and SCT, leads to better ID and OOD local region extraction for SubCoOp. Figure 6 compares the ID/OOD local region extraction selected by LoCoOp and our SubCoOp. The top row shows the original ID images, while the middle and bottom rows correspond to LoCoOp and SubCoOp, respectively. SubCoOp applies subspace regularization to explicitly disentangle ID-relevant and ID-irrelevant features, yielding more coherent and semantically meaningful OOD regions. This results in cleaner ID/OOD separation both visually and quantitatively.

Table 3 presents the hard OOD detection results of our SubCoOp method across multiple ID–OOD dataset pairs. One can note that SubCoOp consistently outperforms the competing baseline SCT in all four cases, highlighting its robustness against semantically hard OOD data.

### 4.3 ABLATION STUDIES

**Performance Enhancement by SR.** As discussed, the critical component of our proposed approach is the subspace regularizations (SR) as defined in equation 6. In this section, we analyze the contribution of each component of the SR in enhancing the OOD detection performance. Table 4 shows that removing both the ID and OOD regularization terms (i.e., setting $\lambda_1 = 0$, $\lambda_2 = 0$) leads to the lowest detection performance among all tested scenarios. Introducing only the ID subspace regularization (i.e., $\lambda_1 \neq 0$, $\lambda_2 = 0$) yields substantial performance improvement, as projecting ID-relevant features onto the subspace spanned by the prompt vectors enhances the desired ID-OOD separability during inference. On the other hand, applying only the OOD subspace regularization provides limited performance gain, as expected. The best results are obtained when both regularization terms are used jointly, underscoring the effectiveness of simultaneously projecting ID features onto the column space and OOD features onto the orthogonal null space.

**Different Image Encoders.** We evaluate our proposed SubCoOp method across different image encoder architectures for CLIP, with results summarized in Fig. 7. The results show that SubCoOp consistently outperforms SCT across ViT-B/16, ViT-B/32 and ResNet (RN)-50 backbones and observed to be particularly effective in transformer-based models. SubCoOp achieved the best performance using the ViT-B/16 architecture. With ViT-B/32 as well, SubCoOp outperforms SCT by reducing the average FPR95 by 0.92% and increasing AUROC by 0.56%. For ResNet-50 architecture, SubCoOp maintains competitive OOD detection performance.

**Varying SR Hyperparameters.** We analyze the impact of varying ID SR hyperparameter $\lambda_1$ and OOD SR hyperparameter $\lambda_2$ (see equation 8) on the performance of our SubCoOp method, as

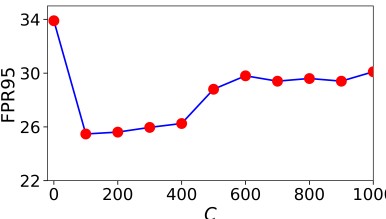

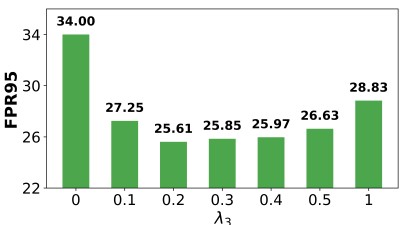

Figure 9: Average OOD detection performance of SubCoOp across different values of $C$ using ImageNet-1k dataset.

Figure 10: Average OOD detection performance of SubCoOp across different values of ER regularizer $\lambda_3$ using ImageNet-1k dataset.

shown in Fig. 8. In general, $\lambda_1 = 2$ achieves the lowest FPR95 and AUROC in ImageNet-1k, while maintaining more or less consistent performance across different $\lambda_2$ values. As one can observe, our method is more sensitive to ID SR regularizer $\lambda_1$. Both excessively high and low values of $\lambda_1$ degrade OOD detection performance. When $\lambda_1$ is small, the regularizer has minimal influence, resulting in unstructured latent features. In contrast, excessive regularization constrains the latent representations too tightly to a learned subspace, potentially suppressing certain discriminative features that are crucial for distinguishing ID from OOD samples. Hence, selecting an appropriate value for $\lambda_1$ is crucial for our approach. As shown in Fig. 8, the configuration $\lambda_1 = 2, \lambda_2 = 5$ achieves the best overall performance, resulting an FPR95 of 25.47% and an AUROC of 94.07%.

**Varying ER Hyperparameter $\lambda_3$ and Rank Threshold $C$.** Fig. 9 and 10 analyzes the impact of varying $C$ values and ER hyperparameter $\lambda_3$, respectively, for our SubCoOp approach. We evaluate $C$ values ranging from 0 to 1,000 under the 16-shot setting. SubCoOp exhibits degraded performance at $C = 0$, where all local regions are treated as OOD, leading to high false positive rates. In Fig. 9, as $C$ increases, particularly in the range of between 100 and 400, FPR95 decreases and AUROC improves, indicating more accurate selection of OOD-relevant local features. In Fig. 10, as the regularization parameter $\lambda_3$ increases from 0, SubCoOp shows performance improvement with a notable decrease in FPR95 and an increase in AUROC, achieving peak performance around weight 0.2. Beyond a weight of 0.3, SubCoOp's performance slightly deteriorates, suggesting that overly strong ER regularization may hinder detection. More ablation studies, implementation settings, and related discussions are presented in supplementary section.

## 5 CONCLUSION

In this work, we propose a novel approach that integrates subspace representation learning with prompt optimization in VLMs for few-shot OOD detection. Our method induces a distinctive geometry in the feature embedding space by projecting ID features onto a subspace spanned by learnable prompt vectors, while pushing ID-irrelevant features to the orthogonal null space. Experiments on several OOD benchmarks based on ImageNet-1k and ImageNet-100 demonstrate that our prompt tuning framework, SubCoOp, consistently outperforms state-of-the-art methods in OOD detection, without sacrificing ID classification accuracy.

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

**Supplementary Material of "Prompt Optimization Meets Subspace Representation Learning for Few-shot Out-of-Distribution Detection"**

## A    RELATED WORKS

**OOD Detection**. Traditional approaches to OOD detection can be broadly categorized into logit-based (Liu et al., 2020b; Hendrycks et al., 2022; Sun et al., 2021), feature-based (Lee et al., 2018; Saito et al., 2020; Park et al., 2023a), probability-based (Sun et al., 2021; Basart et al., 2022; Liang et al., 2017; Huang et al., 2021), and reconstruction-based methods (Yang et al., 2024b). Feature-based methods extract intermediate representations from ID data using a discriminative model and measure distances, such as the Mahalanobis distance, between test samples and the ID feature distribution (Denouden et al., 2018). Recent variants improve robustness by leveraging self-supervised or pre-trained models for more discriminative features (Tack et al., 2020; Sehwag et al., 2021). Recently ViM (Wang et al., 2022) introduces a virtual OOD logit by projecting features onto a residual space and matching it with class logits to compute a robust joint OOD confidence score. GEN (Liu et al., 2023) uses a generalized entropy score computed from the output softmax score, amplifying small deviations from one-hot predictions to separate ID and OOD samples. NNGuide (Park et al., 2023b) uses nearest-neighbor feature similarity to guide confidence and reduce over-confidence on OOD samples.

**Training-Free OOD Detection with Vision-Language Models**. The advent of vision-language models, particularly CLIP (Radford et al., 2021), has opened new research frontiers for training-free OOD detection by leveraging powerful pre-trained joint representations. These methods utilize scoring functions to quantify the semantic discrepancy between ID and OOD samples without requiring model fine-tuning. Early works such as ZOC (Esmaeilpour et al., 2022), and MCM (Ming et al., 2022) apply CLIP-based embeddings for OOD detection using similarity-based metrics. GL-MCM (Miyai et al., 2025) extends this MCM score by incorporating local visual features to enhance OOD detection performance. CLIPN (Wang et al., 2023) proposes negative text encoders to better seperate OOD samples. DPM (Zhang et al., 2024b) matches domain-specific visual features with both textual and visual prototypes, improving ID–OOD separability. TAG (Liu & Zach, 2024) introduces text prompt augmentation strategy to increase the seperation between ID and OOD samples without requiring prompt optimization. In addition, outlier exposure methods like NegLabel (Jiang et al., 2024) leverages a large set of semantically diverse negative labels from WordNet (Fellbaum, 2010) to enhance separability between ID and OOD samples. LAPT (Zhang et al., 2024a) utllizes large text corpora as external knowledge to mine negative label automatically and optimize distribution-aware prompts. CLIP-Scope (Fu et al., 2025) mines nearest and farthest WordNet (Fellbaum, 2010) labels for broad OOD coverage and applies a Bayesian posterior update using historical class-likelihoods to enhance zero-shot OOD detection. Meanwhile, OLE (Ding & Pang, 2024) explore synthetic outlier generation and EOE (Cao et al., 2024) utilizes expert-guided knowledge to improve OOD detection task.

**Prompt Learning for OOD Detection**. Prompt learning has recently emerged as an effective and parameter-efficient paradigm for adapting foundation models to novel tasks under limited supervision. Initially introduced in NLP (Petroni et al., 2019), prompt tuning utllizes trainable prompt tokens to the input rather than updating the full model. In the vision-language domain, CoOp (Zhou et al., 2022b) proposes learning a set of shared context tokens, while CoCoOp (Zhou et al., 2022a) improves adaptability by making prompts conditional on the visual input features. VPT (Jia et al., 2022) further extends this approach by injecting prompts into the visual encoder layers. While this approaches are effective for in-distribution classification, they often struggle in OOD settings, as prompt tuning methods usually optimize for ID accuracy without explicitly addressing semantic shifts in OOD inputs. To address this, LoCoOp (Miyai et al., 2023a) regularizes prompt learning using CLIP's local features as surrogate OOD features. Similarly, LSN (Nie et al., 2024) and Neg-Prompt (Li et al., 2024) incorporate negative prompts to enhance the semantic separation between ID and OOD categories. ID-Like Bai et al. (2024) constructs challenging OOD samples by cropping ID images in the vicinity space and selecting low-similarity regions using CLIP.

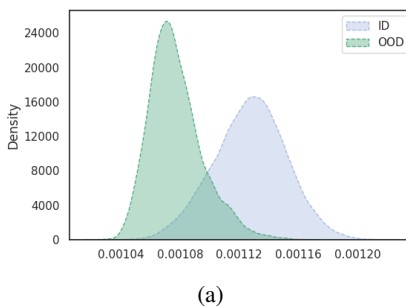 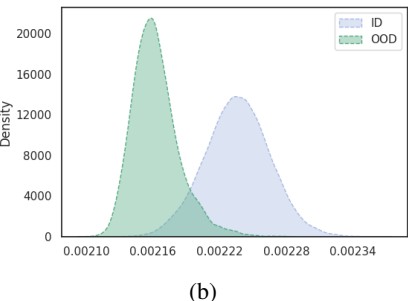

(a)                                                  (b)

Figure 11: OOD detection performance on the iNaturalist dataset using (a) SCT and (b) SubCoOp (Ours).

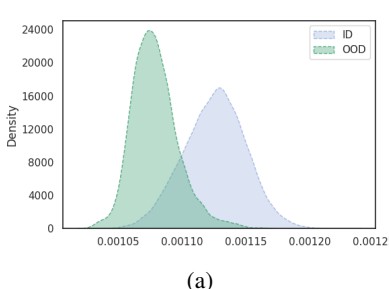 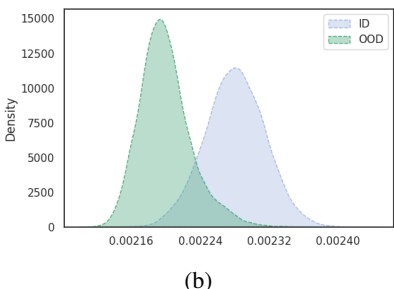

(a)                                                  (b)

Figure 12: OOD detection performance on the SUN dataset using (a) SCT and (b) SubCoOp (Ours).

## B  ADDITIONAL EXPERIMENTS

Figures 11 and 12 show ID/OOD histograms for iNaturalist and SUN, comparing SCT with Sub-CoOp. SubCoOp provides better separation between ID and OOD distributions, with reduced overlap in plots (b), highlighting its stronger capability to discriminate OOD samples. This minimal overlap between ID and OOD distributions demonstrates the strong separability achieved by our subspace regularization.

Table 9 compares the OOD detection performance of SCT and SubCoOp under varying few-shot settings using ImageNet-1k as the ID data set. Both methods demonstrate consistent improvements in FPR95 and AUROC as the number of shots increases. SubCoOp demonstrates a substantial performance gain over SCT, especially in higher-shot settings, achieving lower average FPR95 and higher average AUROC. In the 8-shot setting, SubCoOp consistently outperforms SCT across all OOD datasets, reducing average FPR95 from 30.57% to 26.11% and improving AUROC from 92.94% to 93.49%. Notable gains include substantial FPR95 reductions on SUN of 5.76% and Texture of 7.35%, highlighting SubCoOp's superior OOD separability when more labeled examples are available.

Table 5 presents OOD detection results across four OOD datasets for two backbone architectures, ViT-B/32 and RN-50, comparing SCT with SubCoOp. We present the ID classification performance of different methods in Table 6. Zero-shot and post-hoc methods achieve 66.7% ID accuracy on ImageNet-1k, whereas prompt-tuning approaches such as CoOp and NegPrompt improve this to approximately 71.92%. On the other hand, IDLike and LoCoOp attain 71.04% and 71.43% ID accuracy, respectively. Our proposed SubCoOp achieves a comparable 70.57% ID accuracy while delivering the best overall OOD detection performance.

Table 7 presents the effect of varying the entropy regularization weight ($\lambda_3$) on OOD detection performance for SubCoOp. It is evident that moderate values of $\lambda_3$ between 0.2 and 0.3 consistently obtain the best trade-off between FPR95 and AUROC across all four OOD datasets. For instance, $\lambda_3$=0.2 achieves the lowest average FPR95 of 25.64% with a corresponding AUROC of 94.02%, while $\lambda_3$=0.3 delivers a comparable FPR95 of 25.85% but slightly higher AUROC of 93.98%. Extremely low value $\lambda_3$=0 or high value $\lambda_3$=1.0 results in degraded performance.

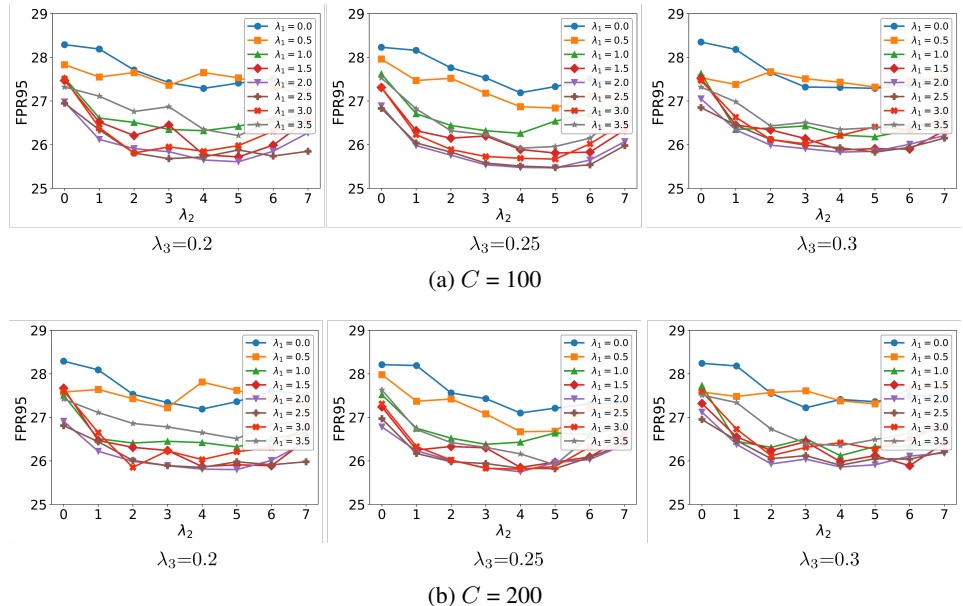

Figure 13: Performance analysis of our proposed SubCoOp method across different hyperparameters using ImageNet-1k dataset.

Table 5: OOD detection performance analysis with different backbones with ImageNet-1k dataset.

| Model | Method | iNaturalist | | SUN | | Places | | Texture | | Avg | |
|---|---|---|---|---|---|---|---|---|---|---|---|
| | | FPR95↓ | AUROC↑ | FPR95↓ | AUROC↑ | FPR95↓ | AUROC↑ | FPR95↓ | AUROC↑ | FPR95↓ | AUROC↑ |
| ViT-B/32 | SCT | $29.07^{\pm3.90}$ | $94.24^{\pm0.44}$ | $35.27^{\pm2.87}$ | $92.47^{\pm0.40}$ | $39.59^{\pm2.40}$ | $90.36^{\pm0.42}$ | $47.13^{\pm1.01}$ | $88.49^{\pm0.80}$ | $37.77^{\pm2.55}$ | $91.39^{\pm0.52}$ |
| | SubCoOp (Ours) | $27.50^{\pm2.24}$ | $94.63^{\pm0.30}$ | $33.85^{\pm0.50}$ | $92.89^{\pm0.27}$ | $38.46^{\pm1.00}$ | $91.42^{\pm0.37}$ | $47.57^{\pm2.29}$ | $88.77^{\pm0.62}$ | $36.85^{\pm1.51}$ | $91.93^{\pm0.39}$ |
| RN-50 | SCT | $40.33^{\pm1.25}$ | $91.83^{\pm0.12}$ | $36.43^{\pm1.10}$ | $91.65^{\pm0.19}$ | $43.78^{\pm1.36}$ | $88.15^{\pm0.43}$ | $37.62^{\pm0.93}$ | $90.36^{\pm0.16}$ | $39.54^{\pm1.16}$ | $90.50^{\pm0.23}$ |
| | SubCoOp (Ours) | $40.09^{\pm1.08}$ | $92.21^{\pm0.14}$ | $36.12^{\pm0.96}$ | $91.74^{\pm0.22}$ | $43.08^{\pm1.41}$ | $88.63^{\pm0.73}$ | $37.51^{\pm0.53}$ | $90.31^{\pm0.13}$ | $39.20^{\pm1.00}$ | $90.72^{\pm0.30}$ |

Figure 13 shows the sensitivity analysis of SubCoOp under different hyperparameter settings. We vary $\lambda_1$ and $\lambda_2$ while fixing $\lambda_3 \in \{0.2, 0.25, 0.3\}$ and report the FPR95 performance for $C = 100$ and $C = 200$. SubCoOp demonstrates stable behavior across a wide range of configurations, with consistent performance near moderate values of $\lambda_1$ and $\lambda_2$. This highlights the robustness of our method to different hyperparameter variations.

Table 8 reports the OOD detection performance of various methods on ImageNet-1k under the 1-shot setting using the CLIP-ViT/B-16 backbone. Among prompt-tuning approaches, SubCoOp achieves the lowest FPR95 of 32.18% and the highest AUROC of 91.83%, outperforming the other state-of-the-art baselines across most OOD datasets. The SR-enhanced variants, CoOp-SR and LoCoOp-SR, improve OOD separability compared to their respective baselines. Specifically, CoOp-SR reduces the average FPR95 from 40.80% for CoOp to 38.31% while maintaining a high AUROC of 89.69% compared to 89.77% for CoOp. These results demonstrate that SubCoOp and SR-based enhancements offer consistent gains in few-shot OOD detection over standard prompt-tuning baselines.

Table 10 presents the OOD detection performance of CoOp, LoCoOp, SCT, and the proposed SubCoOp on ImageNet-100 as ID data. SubCoOp achieves the best overall results, with the lowest average FPR95 of 11.60% and the highest average AUROC of 97.80%, consistently outperforming the other methods across all four OOD datasets. While SCT also delivers strong results with FPR95 12.50% and AUROC 97.25%, SubCoOp offers further gains, demonstrating its robustness and effectiveness in few-shot OOD detection.

Table 6: ID classification accuracy with different baselines(%) utilizing ImageNet-1k dataset

| Method | ID Accuracy |
|---|---|
| *Zero-shot methods* | |
| MCM | 66.7 |
| GL-MCM | 66.7 |
| *CLIP-based post-hoc methods* | |
| MSP | 66.7 |
| ODIN | 66.7 |
| Energy | 66.7 |
| ReAct | 66.7 |
| MaxLogit | 66.7 |
| *Prompt tuning based methods* | |
| CoOp | 71.93 |
| NegPrompt | 71.93 |
| SCT | 71.72 |
| LoCoOp | 71.43 |
| IDlike | 71.04 |
| SubCoOp | 70.57 |

Table 7: OOD detection anslysis utilizing ImageNet-1k as ID dataset with varying entropy regularization weights ( $\lambda_3$ ).

| Method | iNaturalist | | SUN | | Places | | Texture | | Average | |
|---|---|---|---|---|---|---|---|---|---|---|
| | FPR95↓ | AUROC↑ | FPR95↓ | AUROC↑ | FPR95↓ | AUROC↑ | FPR95↓ | AUROC↑ | FPR95↓ | AUROC↑ |
| SubCoOP ($\lambda_3$=0) | 16.20 | 96.67 | 34.36 | 93.60 | 42.19 | 90.09 | 43.19 | 89.43 | 34.00 | 92.45 |
| SubCoOP ($\lambda_3$=0.1) | 12.82 | 97.28 | 18.97 | 95.72 | 29.62 | 92.46 | 41.16 | 90.69 | 25.64 | 94.02 |
| SubCoOP ($\lambda_3$=0.3) | 15.31 | 96.59 | 18.16 | 96.35 | 28.89 | 92.67 | 41.03 | 90.61 | 25.85 | 93.98 |
| SubCoOP ($\lambda_3$=0.4) | 15.61 | 96.58 | 19.14 | 95.82 | 29.38 | 92.12 | 40.60 | 90.51 | 25.97 | 93.78 |
| SubCoOP ($\lambda_3$=0.5) | 14.52 | 97.08 | 19.82 | 95.56 | 29.94 | 91.91 | 42.45 | 90.27 | 26.83 | 93.71 |
| SubCoOP ($\lambda_3$=1.0) | 15.08 | 96.71 | 22.01 | 94.89 | 34.00 | 91.19 | 44.22 | 88.91 | 28.83 | 92.93 |

Table 8: Comparison of FPR95 and AUROC scores on various OOD datasets with ID dataset ImageNet-1k. All methods use the same CLIP-ViT-B/16 backbone, and 1-shot training setting.

| Method | iNaturalist | | SUN | | Places365 | | Textures | | Average | |
|---|---|---|---|---|---|---|---|---|---|---|
| | FPR95↓ | AUROC↑ | FPR95↓ | AUROC↑ | FPR95↓ | AUROC↑ | FPR95↓ | AUROC↑ | FPR95↓ | AUROC↑ |
| | *Prompt tuning based methods (1-shot)* | | | | | | | | | |
| CoOp | $27.99^{\pm4.18}$ | $93.73^{\pm1.27}$ | $36.03^{\pm4.02}$ | $90.95^{\pm0.57}$ | $45.46^{\pm4.26}$ | $87.82^{\pm1.42}$ | $53.70^{\pm1.79}$ | $84.59^{\pm0.67}$ | $40.80^{\pm3.56}$ | $89.77^{\pm0.98}$ |
| LoCoOp | $26.81^{\pm2.78}$ | $94.45^{\pm0.72}$ | $26.16^{\pm1.13}$ | $94.06^{\pm0.21}$ | $35.18^{\pm1.05}$ | $91.10^{\pm0.13}$ | $50.53^{\pm0.33}$ | $86.96^{\pm0.60}$ | $34.67^{\pm1.32}$ | $91.64^{\pm0.42}$ |
| IDLike | $12.07^{\pm0.88}$ | $97.65^{\pm0.10}$ | $40.55^{\pm5.84}$ | $91.07^{\pm1.80}$ | $47.94^{\pm5.24}$ | $88.31^{\pm2.05}$ | $38.34^{\pm13.39}$ | $89.67^{\pm4.03}$ | $34.72^{\pm0.80}$ | $91.67^{\pm0.07}$ |
| NegPrompt | $65.03^{\pm8.69}$ | $84.56^{\pm2.52}$ | $44.39^{\pm1.66}$ | $89.63^{\pm0.66}$ | $51.31^{\pm6.21}$ | $86.55^{\pm2.19}$ | $63.76^{\pm3.02}$ | $83.76^{\pm3.02}$ | $62.08^{\pm3.71}$ | $81.13^{\pm1.78}$ |
| LSN | $59.28^{\pm7.02}$ | $87.20^{\pm3.15}$ | $40.15^{\pm0.82}$ | $91.47^{\pm0.14}$ | $46.11^{\pm1.86}$ | $88.74^{\pm0.57}$ | $60.34^{\pm0.14}$ | $88.92^{\pm0.42}$ | $51.47^{\pm1.53}$ | $87.84^{\pm0.58}$ |
| SCT | $20.77^{\pm4.12}$ | $95.15^{\pm1.15}$ | $24.92^{\pm2.03}$ | $94.17^{\pm0.53}$ | $33.35^{\pm2.03}$ | $91.08^{\pm0.49}$ | $50.28^{\pm1.18}$ | $85.71^{\pm0.08}$ | $32.83^{\pm2.34}$ | $91.53^{\pm0.56}$ |
| **SubCoOp** | $20.44^{\pm4.71}$ | $94.96^{\pm1.01}$ | $24.13^{\pm3.60}$ | $94.36^{\pm0.92}$ | $32.45^{\pm3.02}$ | $91.78^{\pm0.72}$ | $50.07^{\pm1.11}$ | $87.15^{\pm0.08}$ | $31.96^{\pm3.11}$ | $91.83^{\pm0.68}$ |
| **CoOp-SR** | $23.87^{\pm4.49}$ | $94.88^{\pm1.32}$ | $34.68^{\pm1.65}$ | $90.72^{\pm0.74}$ | $42.86^{\pm1.14}$ | $88.92^{\pm0.71}$ | $51.83^{\pm1.67}$ | $84.25^{\pm0.31}$ | $38.31^{\pm2.24}$ | $89.69^{\pm0.77}$ |
| **LoCoOp-SR** | $26.31^{\pm7.74}$ | $94.47^{\pm1.43}$ | $26.13^{\pm1.89}$ | $94.41^{\pm0.23}$ | $34.85^{\pm1.14}$ | $91.26^{\pm0.28}$ | $50.62^{\pm1.67}$ | $87.01^{\pm0.31}$ | $34.48^{\pm3.11}$ | $91.79^{\pm0.56}$ |

Table 9: Comparison of FPR95 and AUROC scores using different few-shot techniques (%) on various OOD datasets with ID dataset ImageNet-1k.

| Method | iNaturalist | | SUN | | Places | | Texture | | Average | |
|---|---|---|---|---|---|---|---|---|---|---|
| | FPR95↓ | AUROC↑ | FPR95↓ | AUROC↑ | FPR95↓ | AUROC↑ | FPR95↓ | AUROC↑ | FPR95↓ | AUROC↑ |
| SCT (1-shot) | $20.77^{\pm4.12}$ | $95.15^{\pm1.15}$ | $24.92^{\pm2.03}$ | $94.17^{\pm0.53}$ | $33.35^{\pm2.03}$ | $91.08^{\pm0.49}$ | $50.28^{\pm1.18}$ | $85.71^{\pm0.08}$ | $32.83^{\pm2.34}$ | $91.53^{\pm0.56}$ |
| SCT (4-shot) | $22.78^{\pm1.06}$ | $95.01^{\pm0.56}$ | $22.97^{\pm0.72}$ | $95.16^{\pm0.41}$ | $33.10^{\pm2.01}$ | $91.80^{\pm0.43}$ | $44.68^{\pm2.22}$ | $89.12^{\pm0.68}$ | $30.88^{\pm1.50}$ | $92.77^{\pm0.52}$ |
| SCT (8-shot) | $17.45^{\pm1.19}$ | $96.50^{\pm0.20}$ | $24.23^{\pm0.23}$ | $94.82^{\pm0.31}$ | $33.90^{\pm0.58}$ | $91.69^{\pm0.19}$ | $46.71^{\pm2.09}$ | $88.74^{\pm0.67}$ | $30.57^{\pm1.02}$ | $92.94^{\pm0.34}$ |
| SCT (16-shot) | $16.14^{\pm1.81}$ | $96.68^{\pm0.29}$ | $21.57^{\pm1.20}$ | $95.23^{\pm0.26}$ | $31.47^{\pm0.89}$ | $91.89^{\pm0.25}$ | $43.75^{\pm0.56}$ | $88.83^{\pm0.45}$ | $28.23^{\pm1.12}$ | $93.16^{\pm0.31}$ |
| SubCoOp (1-shot) | $20.44^{\pm4.71}$ | $94.96^{\pm1.01}$ | $24.13^{\pm3.60}$ | $94.36^{\pm0.92}$ | $32.45^{\pm3.02}$ | $91.78^{\pm0.72}$ | $50.07^{\pm1.11}$ | $87.15^{\pm0.08}$ | $31.96^{\pm3.11}$ | $91.83^{\pm0.68}$ |
| SubCoOp (4-shot) | $15.16^{\pm2.58}$ | $96.62^{\pm0.43}$ | $19.55^{\pm2.06}$ | $95.71^{\pm0.57}$ | $29.09^{\pm0.91}$ | $92.54^{\pm0.23}$ | $44.06^{\pm1.68}$ | $89.73^{\pm0.69}$ | $26.96^{\pm1.81}$ | $93.61^{\pm0.48}$ |
| SubCoOp (8-shot) | $14.65^{\pm2.62}$ | $96.48^{\pm0.32}$ | $18.47^{\pm0.34}$ | $95.88^{\pm0.07}$ | $27.28^{\pm4.91}$ | $93.06^{\pm1.62}$ | $43.22^{\pm1.22}$ | $89.76^{\pm0.14}$ | $25.91^{\pm2.27}$ | $93.80^{\pm0.54}$ |
| SubCoOp (16-shot) | $12.61^{\pm1.69}$ | $97.28^{\pm0.38}$ | $18.75^{\pm1.47}$ | $95.82^{\pm0.20}$ | $29.45^{\pm1.66}$ | $92.51^{\pm0.13}$ | $41.06^{\pm1.02}$ | $90.65^{\pm0.25}$ | $25.47^{\pm1.46}$ | $94.07^{\pm0.24}$ |

Table 10: Comparison of FPR95 and AUROC scores (%) on various OOD datasets with ID dataset ImageNet-100.

| Method | iNaturalist | | SUN | | Places | | Texture | | Average | |
|---|---|---|---|---|---|---|---|---|---|---|
| | FPR95↓ | AUROC↑ | FPR95↓ | AUROC↑ | FPR95↓ | AUROC↑ | FPR95↓ | AUROC↑ | FPR95↓ | AUROC↑ |
| CoOp | $23.70^{\pm6.29}$ | $96.67^{\pm0.57}$ | $21.30^{\pm6.00}$ | $96.53^{\pm0.51}$ | $25.75^{\pm2.37}$ | $95.28^{\pm0.42}$ | $19.39^{\pm1.27}$ | $96.85^{\pm0.16}$ | $22.54^{\pm3.98}$ | $96.33^{\pm0.42}$ |
| LoCoOp | $11.30^{\pm10.01}$ | $97.99^{\pm0.46}$ | $13.90^{\pm7.35}$ | $96.92^{\pm0.29}$ | $20.57^{\pm10.13}$ | $95.50^{\pm0.39}$ | $17.23^{\pm8.56}$ | $96.16^{\pm0.52}$ | $15.75^{\pm9.01}$ | $96.64^{\pm0.42}$ |
| SCT | $5.26^{\pm0.21}$ | $98.71^{\pm0.35}$ | $11.21^{\pm3.20}$ | $97.54^{\pm1.16}$ | $16.21^{\pm4.12}$ | $96.47^{\pm0.78}$ | $17.32^{\pm1.76}$ | $96.29^{\pm0.63}$ | $12.50^{\pm2.32}$ | $97.25^{\pm0.81}$ |
| SubCoOp | $5.03^{\pm1.93}$ | $98.83^{\pm0.28}$ | $9.70^{\pm0.98}$ | $98.03^{\pm0.23}$ | $15.06^{\pm0.32}$ | $96.73^{\pm0.07}$ | $16.59^{\pm0.58}$ | $97.59^{\pm0.31}$ | $11.60^{\pm0.95}$ | $97.80^{\pm0.22}$ |

