# OpenReview forum: "Prompt Optimization Meets Subspace Representation Learning for Few-shot Out-of-Distribution Detection"
_ICLR.cc/2026/Conference — Submitted to ICLR 2026_

### Official Review · Reviewer_yiNu · 2025-10-20

**Soundness:** 3
**Presentation:** 3
**Contribution:** 3
**Rating:** 6
**Confidence:** 5

**Summary:**

This paper tackles the problem of few-shot semantic OOD detection. It extends the LoCoOP framework with an additional loss term $\mathcal{L}_\text{sub}$. Specifically, after obtaining the pseudo-labels of local features obtained by CLIP embeddings, it projects the ID-relevant features to the subspace of  learnable prompt matrix $W$, and projects the OOD features to the orthogonal complement space.  The proposed method is easy to implement and effective.

**Strengths:**

Strengths:

-  The paper is well-written and easy to follow.
-  The proposed loss term is technically sound and intuitive.
-  The experiments are extensive.

**Weaknesses:**

Weaknesses:
-   Several recent works are missing.
    - Designing a scoring function has been a central topic in the community, it is a bit disrespectful not to cite recent works on this direction. Particularly, ViM [1], GEN[2], and NN-Guide[3] are three strong baselines. Meanwhile, it is not completely true that zero shot OOD detection "depends heavily on manually crafted prompts, where even slight variations (e.g., “a flower” vs. “a type of a flower”) can significantly impact performance". Indstead, TAG [4] takes the advantage of this phenomena to enhance the performance of OOD detection, and NegLabel [5] only requires OOD labels. Neither of them are sensitive to prompt templates or require fine-tuning.
-   The experimental settings do not follow the standard OOD benchmark (https://zjysteven.github.io/OpenOOD/).
     - The authors are expected to include the results on OpenImage-O.

- The experiments are only done for ViT-B/16.
   - The authors are expected to show results on ResNet-50.

[1] ViM: Out-Of-Distribution with Virtual-logit Matching. CVPR, 2022.
[2] GEN: Pushing the Limits of Softmax-Based Out-of-Distribution Detection. CVPR, 2023.
[3] Nearest Neighbor Guidance for Out-of-Distribution Detection. ICCV, 2023.
[4] TAG: Text Prompt Augmentation for Zero-Shot Out-of-Distribution Detection. ECCV, 2024.
[5] Negative Label Guided OOD Detection with Pretrained Vision-Language Models. ICLR, 2024.

**Questions:**

See weaknesses.

---

> ### Author Response · Authors · 2025-11-23
> **Response to Reviewer yiNu**
>
> We would like to thank the reviewer for your reviewing efforts and insightful comments. Please see our detailed response as follows:
>
> **[Discussion on additional baselines]** We appreciate the reviewer’s comment and completely agree with adding more baselines in the related works section. There exists a large body of related works in OOD domain. Due to page limitations, in the main text we focused our discussion on the most closely related methods, especially prompt-learning-based approaches. **A more detailed discussion of related work was provided in the supplementary material, where we describe several categories of OOD methods, including some of the reviewer’s suggested methods (e.g., NegLabel)**. In the revised manuscript, we have explicitly discussed and cited the suggested strong baselines, including ViM, GEN, and NN-Guide.
>
>
>
> **[Sensitivity of manually crafted prompts]** For CLIP-based zero-shot classification, CoOp [1], which introduced the prompt-learning framework, demonstrates that performance can be sensitive to minor variations in text prompts (see Fig. 1 in [1]). TAG proposes a text-augmentation strategy that effectively ensembles multiple prompts to mitigate this sensitivity to fixed text prompts. Prompt-learning methods such as CoOp, LoCoOp, SCT, and our SubCoOp explicitly optimize the prompts for the target scenario rather than relying on hand-crafted prompts. In contrast, NegLabel leverages external resources such as WordNet, whereas approaches like TAG and SubCoOp do not require any external knowledge bases. In our revised manuscript, we have added discussions to clarify these distinctions.
>
> [1] Zhou, Kaiyang, Jingkang Yang, Chen Change Loy, and Ziwei Liu. "Learning to prompt for vision-language models." International Journal of Computer Vision 130, no. 9 (2022): 2337-2348.
>
>
>
> **[OpenImage-O]** We thank the reviewer for this comment. As suggested by the reviewer, we have now included OOD detection performance on OpenImage-O dataset as well.
>
> Table1: OOD detection performance with OpenImage-O dataset as OOD and ImageNet-1k as ID. SubCoOp remains competitive compared to other prompt-learning approaches
>
> | Method |  FPR95  | AUROC |
> |--------|------:|------:|
> | SubCoOp | **29.14** | **93.52** |
> | SCT     | 32.62 | 92.78 |
> | LoCoOp  | 34.37 | 92.25 |
> | CoOp    | 35.85 | 91.92 |
>
> **[Experiments with ResNet-50 archiecture]** We have already reported results with multiple backbones, including ResNet-50 and ViT-B/32. Each setting was evaluated over three runs, and the averaged performance is presented in Figure 7 and Table 5.

---

> > ### Comment · Reviewer_yiNu · 2025-11-27
> > **Thanks for conducting the suggested experiments and corresponding revision.**
> >
> > I do not have further questions, and am willing to maintain my score.

---

> > > ### Author Response · Authors · 2025-11-27
> > >
> > > Thank you for acknowledging our work.

---

### Official Review · Reviewer_KQBt · 2025-10-29

**Soundness:** 2
**Presentation:** 2
**Contribution:** 2
**Rating:** 2
**Confidence:** 4

**Summary:**

This paper focuses on enhancing the OOD detection capability of VLMs under the setting of few-shot training with only ID data. It proposes a new learning framework which projects the ID features into a subspace spanned by the prompt vectors, while simultaneously projecting ID-irrelevant features into the orthogonal null space. Experiments on ImageNet OOD benchmarks demonstrate that the proposed method outperforms state-of-the-art few-shot OOD detection approaches.

**Strengths:**

1. The proposed method is clearly presented and easy to understand.
2. Various ablation experiments are conducted to analyze the proposed method.

**Weaknesses:**

1.  **The motivation of the proposed method is logically weak and flawed**. The paper claims that existing prompt learning-based OOD methods overlook the geometry of the visual feature embeddings learned by VLMs, but the reviewer believes that the loss function of LoCoOp inherently incorporates the geometry of the visual features. To be specific, the cross-entropy loss is defined as:

$$
\mathcal{L}\_{CE}=-\log\frac{exp(sim(f^{in},g\_k)/ \tau)}{\sum\_{k'=1}^K exp(sim(f^{in},g\_{k'})/ \tau)}=-(\frac{sim(f^{in},g\_k)}{\tau}-\log(\sum\_{k'=1}^K \frac{exp(sim(f^{in},g\_{k'}))}{\tau}))
$$

where the ground truth label $y=k$ and $sim(f^{in},g_k)$ represents the cosine value of the angle between visual features $f^{in}$ and textual features $g_k$. Minimizing $\mathcal{L}_{CE}$ is basically equivalent to maximizing the cosine similarity between $f^{in}$ and $g_k$ and minimizing the cosine similarity between $f^{in}$ and the textual features of other ID classes, which indeed involves the geometry of the visual features. If we consider the subspace spanned by the textual features of all ID classes, i.e., $W=[g_1, g_2, \dots, g_K]$, the original framework of LoCoOp can also be seen as a form of subspace representation learning.

2. **The proposed method seems misaligned with the core design of CLIP**. To be specific, CLIP is pretrained to directly align the visual features (the output of the image encoder) and textual features (the output of the text encoder), rather than the visual features and the embedding of the input prompts (the input of the text encoder). However, the proposed method projects the visual features into the subspace spanned by the learnable prompt vectors. The reviewer suggests that the authors provide an in-depth theoretical analysis or discussion on this misalignment problem.

3. The reviewer **doubts whether it’s reasonable to choose the learnable prompt vectors as the basis of the ID subspace.** The learnable prompt vectors are originally designed to represent the template prompts, such as “This is a photo of”, which contains no ID information at all. The ID information of the textual features is supposed to lie in the ID classnames, rather than the template prompts.  Therefore, using the prompt vectors as the basis for the ID subspace lacks a theoretical justification.

4. **The contribution of this paper is rather empirical.** The authors provide no theoretical analysis or justification for the proposed methods, so extensive experiments are needed to demonstrate that the proposed method can significantly benefit existing methods. But the proposed method **shows marginal improvement on some benchmarks**, such as ImageNet-100 OOD benchmark and using ViT-B/32 and RN-50 as image encoder, with improvement on FPR95 less than 1.

5. **This paper omits discussion on several important related works**, including ID-like [1], NegPrompt [2] and Local-Prompt [3], which are all prompt-tuning-based OOD detection methods. What’s more, ID-like and Local-Prompt both utilize the ID-irrelevant regions for OOD regularization, which are similar to the proposed method.

6. The introduction section presents the representative LoCoOp and SCT methods but **fails to clearly explain the core motivation** of the proposed method. The reviewer suggests that the authors add a more detailed explanation of the motivation in the “Our contributions” paragraph which starts at Line 92.

7. The reviewer suggests that the authors conduct experiments to **evaluate and compare the actual time cost** of a training iteration of LoCoOp and the proposed method, to demonstrate the computation efficiency of the proposed method.

8. Could the authors evaluate the proposed method on the **CIFAR-100 OOD benchmark** and the **hard OOD setting** with ImageNet-10 as ID and ImageNet-100 as OOD?

9. The statement about efficiently extracting more informative proxy-OOD supervision in Line 94 is **confusing and unclear**. Could the authors explain what “extracting more informative proxy-OOD supervision” means? After all, the proposed method uses the same OOD Local Features Extraction as LoCoOp.

10. **The organization of the experiment section could be improved for better clarity and readability.** For instance, the experiment setups, main experiments and ablation study can be presented as three subsections with second-level headings.

[1]. Yichen Bai, et al. ID-like Prompt Learning for Few-Shot Out-of-Distribution Detection

[2]. Tianqi Li, et al. Learning Transferable Negative Prompts for Out-of-Distribution Detection

[3]. Fanhu Zeng, et al. Local-Prompt: Extensible Local Prompts for Few-Shot Out-of-Distribution Detection

**Questions:**

Please see the weaknesses.

**Details Of Ethics Concerns:**

The reviewer does not notice any ethical issues with this paper.

---

> ### Author Response · Authors · 2025-11-23
> **Response to Reviewer KQBT**
>
> We would like to thank the reviewer for careful reading and insightful comments. Our detailed responses are presented as follows.
>
> **[Geometry awareness of the CE loss function in LoCoOp/CoOp vs our method]** We thank the reviewer for this insightful discussion and careful evaluation of our work. Indeed, we agree with the reviewer's interpretation of how the cosine similarities between visual features and textual prompts in the CE loss function leverages pairwise angular geometry between$\boldsymbol{f}^{\text{in}}$ and $\boldsymbol{g_k}$ for all class $k$. However, our claim is that this objective does not explicitly learn to distinguish between the global ID subspace and OOD directions, which is central to our method. Our subspace-based formulation is designed to exploit a discriminative geometry between ID and OOD samples, rather than only shaping the relative geometry among ID classes (which the CE loss already handles, as the reviewer rightly points out). Figure 5 in the revised manuscript illustrates this where ID features tend to form a low-dimensional, well-clustered subspace, whereas OOD features are more dispersed and exhibit a higher-dimensional structure. Nonetheless, we agree that the phrasing “existing prompt-learning–based OOD methods overlook the geometry of the visual feature embeddings learned by VLMs” can be confusing. In the revised manuscript, we have updated this statement to clarify that these methods do not explicitly model the global ID subspace versus OOD directions, in line with the reviewer’s observation.
>
> **[Misalignment with the core design of CLIP]** We thank the reviewer for raising this concern and we would like to clarify that our method does not change CLIP’s core alignment objective. In all our experiments, predictions are obtained exactly as in CLIP/CoOp, i.e., we compute cosine similarities between the visual features $\boldsymbol{f}^{\text{in}}$ from the image encoder and the text features  ${\boldsymbol{g}}_k = \boldsymbol{g}(\{\boldsymbol{\omega}_1, \boldsymbol{\omega}_2, \ldots, \boldsymbol{\omega}_M, \boldsymbol{c}_k\}) = $ from the output of the text encoder, and use these as scores for classification/OOD detection. The subspace projections using $\boldsymbol{W}= [\boldsymbol{\omega}_1, \boldsymbol{\omega}_2, \ldots, \boldsymbol{\omega}_M]$ appear as regularizers in the overall training loss to promote disntinctive geometry of the ID and OOD samples. In fact, the $\boldsymbol{W}$-subspace is a parameter-efficient surrogate for the low-dimensional ID feature space (please also refer to Fig. 5 in the revised manuscript).  Our consistent empirical performance gain also shows that the CLIP's core alignment is not harmed, but rather enhanced for OOD detection.
>
> **[Clarification on choosing learnable prompt vectors as the basis of the ID subspace]** Thank you for raising this point. As the reviewer pointed out, the learnable prompt vectors in existing methods may not be well aligned with ID information. Our goal is to inject ID/OOD geometry awareness into these prompts by aligning them more closely with ID features, i.e., by treating the prompts as ``characteristic" vectors of the ID subspace and learning them through few-shot training. This geometric alignment also explains why LoCoOp and SubCoOp exhibit different ID/OOD region extraction behavior—SubCoOp is often more accurate as its prompts are guided by the ID/OOD subspace structure **(see Fig. 6 in the revised manuscript)**.

---

> ### Author Response · Authors · 2025-11-23
> **Response to Reviewer KQBT**
>
> **[Empirical advantage on ImageNet-100]** We appreciate the reviewer’s concern. We would first like to emphasize that our approach provides substantial performance gains on the larger ImageNet-1K setting, as shown in our main tables. On the smaller ImageNet-100 benchmark, the improvement in FPR95 is indeed around 0.9. However, a key observation is the consistency and stability of our method compared to existing approaches.
>
> Table 1: Comparison of OOD detection performance with 16-shot setting using ImageNet-100 dataset.
> | Method  | Avg FPR95 ↓ | Avg AUROC ↑ |
> |---------|-------------|-------------|
> | CoOp    | 22.54 ± 3.98 | 96.33 ± 0.42 |
> | LoCoOp  | 15.75 ± 9.01 | 96.64 ± 0.42 |
> | SCT     | 12.50 ± 2.32 | 97.25 ± 0.81 |
> | **SubCoOp** | **11.60 ± 0.95** | **97.80 ± 0.22** |
>
>
> For example, SCT and LoCoOp exhibit noticeable variation in their results across different settings, whereas our method is much more stable, so the observed margin is not trivial in practice.
>
> In addition, we report the 4-shot results on the ImageNet-100 dataset, where SubCoOp consistently outperforms both LoCoOp and SCT by a substantial margin.
>
> Table 2: Performance analysis of ImageNet-100 dataset with our SubCoOp method with 4-shot setting.
> | **Method**              | **FPR95** | **AUROC** |
> |------------------------|---------|-----------|
> | LoCoOp                 | 16.56   | 96.68     |
> | SCT                    | 17.32   | 96.56     |
> | SubCoOp           | **14.85** | **97.27** |
>
>
>
> Besides, we have included more benchmark datasets, including OpenImageOOD now to validate our method under a broader range of scenarios.
>
> Table 3: OOD Detection Performance on Openinage-O dataset with ImageNet-1k as ID dataset.
> | Method |  FPR  | AUROC |
> |--------|------:|------:|
> | CoOp    | 35.85 | 91.92 |
> | LoCoOp  | 34.37 | 92.25 |
> | SCT     | 32.62 | 92.78 |
> | SubCoOp | **29.14** | **93.52** |
>
>
>
> **[More discussion about baselines]** Thank you for pointing this out. As suggested by the reviewer, we also include a brief discussion of these methods in the introduction. A more detailed discussion of related work was moved to the supplementary material due to page limitations.
>
> **[Comparison of runtime]** To compare the runtime of different prompt-learning methods, we have added a table reporting the training time of each approach. In the original submission, we already discussed the computational overhead introduced by our subspace regularization in Remark 1, showing that it is dominated by $\mathcal{O}(dM)$ where $d$ is the penultimate feature dimesnion and $M$ is the number prompts. **This overhead is negligible compared to the backbone forward pass, which is also reflected in the table below.**
>
> Table 4: Training time for different prompt learning-based OOD approaches. SubCoOp has an efficient design via utilizing the subspace geometry of the visual features, especially compared to data augmentation-based recent approaches such as OSPCoOp and Local-Prompt. We have used Tesla V100 GPU (32GB) for our experiments.
>
> | Method       | ID/OOD Data Aug. | Training time/epoch | Total training time | Averge FPR95 |
> |:------------:|:----------------:|:-------------------:|:-------------------:|:-------------------:|
> | CoOp         | $\times$         | 0.52 s              | 5 h 30 min          | 37.27 |
> | LoCoOp       | $\times$         | 0.66 s              | 6 h 10 min          | 30.02 |
> | SCT          | $\times$         | 0.66 s              | 3 h 10 min          | 28.23 |
> | SubCoOp      | $\times$         | 0.67 s              | 3 h 15 min          | 25.47 |
> | OSPCoOp       | $\checkmark$     | 2.6 s               | 11 h 38 min         | 25.54 |
> | Local-Prompt | $\checkmark$     | 6.1 s               | 24 h 5 min         | 28.27 |

---

> ### Author Response · Authors · 2025-11-23
> **Response to Reviewer KQBT**
>
> **[More OOD Benchmark datasets]** Thank you for this suggestion. We have already included some hard OOD setting scenarios in Table 3 of our the original submission. We are presenting more results as suggested by the reviewer as follows.
>
> Table 5: OOD detection performance with ImageNet-10 as ID and ImageNet-100 as OOD
> | ID Dataset | OOD Dataset | Method  |  FPR95 | AUROC |
> |-----------|-------------|---------|-----:|------:|
> | ImageNet-10 | ImageNet-100 | SCT     | 6.42 | 97.75 |
> | ImageNet-10 | ImageNet-100 | SubCoOp | **5.92** | **97.95** |
>
> In addition, we are also expanding our analysis to more datasets including OpenImage-O benchmark as follows:
>
> Table 6: OOD Detection Performance on Openinage-O dataset with ImageNet-1k as ID dataset
> | Method |  FPR  | AUROC |
> |--------|------:|------:|
> | CoOp    | 35.85 | 91.92 |
> | LoCoOp  | 34.37 | 92.25 |
> | SCT     | 32.62 | 92.78 |
> | SubCoOp | **29.14** | **93.52** |
>
> **[SubCoOp improves OOD local region extraction]** We would like to respectfully point out the distinction here and clarify this aspect. The OOD local region selection is updated at every epoch during training time in these prompt learning frameworks. As the prompt vectors are better aligned with more informative ID-OOD geometry under the proposed SubCoOp, this eventually helps in better local OOD region extraction compared to what LoCoOp can offer. This advantage is reflected in the improved OOD detection performance as reported in Table I. To further substantiate this claim, we have added a visual analysis comparing the OOD region extraction of LoCoOp and SubCoOp, which clearly illustrates the superiority of our approach (see Fig. 6 on page 8 of the revised manuscript). We have revised the sentence around Line 94 to reflect this discussion as well as added detailed explanation about this around Fig. 5 of the experiment section.
>
> **[Organization of the experiment section]** We appreciate the reviewer’s feedback here. In our original submission, we adopted this organization due to page limitation. Now, we have updated the organization of this section as suggested by the reviewer.

---

### Official Review · Reviewer_J8YY · 2025-10-30

**Soundness:** 3
**Presentation:** 2
**Contribution:** 2
**Rating:** 4
**Confidence:** 4

**Summary:**

This paper proposes SubCoOp, a geometry-aware prompt optimization framework for few-shot OOD detection with CLIP. Prompt vectors are used as a basis for an ID subspace; ID features are encouraged to lie in the column space, while proxy-OOD (local irrelevant) features are pushed into the orthogonal null space via subspace regularization. The loss combines CE, entropy maximization, and two projection-based terms, weighted by model confidence. The method is lightweight, end-to-end, and integrates seamlessly with prior prompt-tuning OOD detectors.

**Strengths:**

- Casting prompts as subspace basis is principled, and parameter-efficient. Orthogonal projections rigorously separate ID and proxy-OOD signals in latent space, complementing softmax-based methods.

- The empirical evaluation confirms that the proposed methodology achieves SOTA performance, consistently outperforming established baselines. Furthermore, the ablation studies provide a clear validation for the contribution of each component within the objective function, justifying the design choices.

**Weaknesses:**

- The novelty of the proposed methodology is questionable. The 'OOD Local Features Extraction' module appears to be a direct application, if not identical to, the LoCoOP method. Concurrently, the reliance on subspace projection constitutes a standard approach that has been extensively explored in prior OOD research [1, 2, 3]. As such, the work feels derivative, and its unique contribution remains unclear.

- The analysis of the learned subspace is underdeveloped. The manuscript would be significantly strengthened by a more rigorous evaluation, including: (1) A sensitivity analysis or ablation study on the subspace dimensionality, $M$, to justify its choice. (2) A more compelling demonstration of the subspace's effectiveness. The empirical results show only a marginal performance gain over the baseline in the original feature space (94.07% vs 93.16% AUROC). This limited improvement raises questions about the practical utility and necessity of the projection component. The authors are encouraged to explore potential refinements, such as enforcing constraints (e.g., basis orthogonality) on the subspace, which might yield a more discriminative representation.

Reference:

[1]: NECO: NEural Collapse Based Out-of-distribution detection

[2]: Out-of-Distribution Detection Using Union of 1-Dimensional Subspaces

[3]: Out-of-distribution detection based on subspace projection of high-dimensional features output by the last convolutional layer

**Questions:**

Please see Weaknesses.

---

> ### Author Response · Authors · 2025-11-23
> **Response to Reviewer J8YY**
>
> We thank the reviewer for appreciating our idea and for providing constructive feedback. Please see our detailed response as follows:
>
> **[Clarification on novelty and contributions]** We thank the reviewer for raising this concern and allowing us to clarify. Although subspace projection ideas have been previously explored in traditional OOD works, our key contribution here is to **systematically integrate subspace geometry into prompt-learning–based OOD frameworks, which to the best of our knowledge has not been studied before**. We build on a LoCoOp-style design as it provides a simple, effective, and computationally efficient way to obtain pseudo-OOD regions. Our goal is not to re-invent this component, but to show that the **effectiveness of such pseudo-OOD region extraction can be substantially improved by utilizing geometry of the ID-OOD features**, leading to enhanced OOD detection via a simple, computationally efficient design. This is a nontrivial shift from the excising prompt learning approaches such as CoOp, LoCoOP, SCT, OSPCoOP, and Local Prompt that rely solely on the predicted confidence scores (CLIP confidence scores have known limitations here) where our geometry-inspired SubCoOp framework complements it well. This is validated in our experiments showing **enhancement brought by SubCoOp across 4 different recent prompt learning OOD approaches (not limited to LoCoOp only)**, demonstrating that the proposed subspace formulation is not merely derivative but provides a principled framework that can consistently strengthens existing methods. We will make these points clearer in the revised manuscript. Also, we present the following table further supporting our clarifications.
>
>
> Table 1: Training time for different prompt learning-based OOD approaches. SubCoOp has an efficient design via utilizing the subspace geometry of the visual features, especially compared to data augmentation-based recent approaches such as OSPCoOp and Local-Prompt. We have used NVIDIA Tesla V100 GPU (32GB) for our experiments.
>
> | Method       | ID/OOD Data Aug. | Training time/epoch | Total training time | Averge FPR95 |
> |:------------:|:----------------:|:-------------------:|:-------------------:|:-------------------:|
> | CoOp         | $\times$        | 0.52 s              | 5 h 30 min          | 37.27 |
> | LoCoOp       | $\times$         | 0.66 s              | 6 h 10 min          | 30.02 |
> | SCT          | $\times$         | 0.66 s              | 3 h 10 min          | 28.23 |
> | SubCoOp      | $\times$         | 0.67 s              | 3 h 15 min          | 25.47 |
> | OSPCoOp       | $\checkmark$     | 2.6 s               | 11 h 38 min         | 25.54 |
> | Local-Prompt | $\checkmark$     | 6.1 s               | 24 h 5 min         | 28.27 |
>
>
> **[More ablation study for SubCoOp]** We thank the reviewer for these comments. Following the reviewer’s suggestion, we have broadened our hyperparameter analysis to include larger ranges and more parameters, as shown in the tables below.
>
>
>
> Table 2: Ablation study on different number of learnable prompts ($M$) on both SCT and SubCoOp
>
> | $M$ | SCT FPR | SCT AUROC | SubCoOp FPR | SubCoOp AUROC |
> |:--------------------:|:--------:|:----------:|:------------:|:--------------:|
> | 4                  |  28.16  |    93.06  |     27.68   |       93.63   |
> | 8                  |  28.55  |    93.00  |     26.72   |       93.64   |
> | 16                 |  28.23  |    93.16  |     25.47   |       94.07   |
> | 24                 |  28.08  |    93.52  |     26.33   |       93.91   |
> | 32                 |  28.12  |    93.47  |     26.85   |       93.61   |
>
> We find that choosing $M=16$ is a reasonable setting, in line with other prompt-learning methods, and that further increasing $M$ yields little gains.
> In addition, we also expand our hyperparameter study of $\lambda_1, \lambda_2$ on larger ranges of values as shown below:
>
>
>
> Table 3: Average detection performance of SubCoOp method in FPR with $\lambda_1$ and $\lambda_2$ using ImageNet-1k as ID dataset.
>
> | $\lambda_1$/ $\lambda_2$ | 0     | 1     | 2     | 3     | 4     | 5     | 6     | 7     |
> |:-----------:|:-------:|:-------:|-------:|:-------:|:-------:|:-------:|:-------:|:-------:|
> | 0.0    | 28.23 | 28.16 | 27.76 | 27.53 | 27.19 | 27.33 | 27.51 | 27.32 |
> | 0.5    | 27.96 | 27.47 | 27.52 | 27.18 | 26.87 | 26.84 | 27.12 | 27.05 |
> | 1.0    | 27.62 | 26.71 | 26.44 | 26.32 | 26.26 | 26.54 | 26.71 | 26.86 |
> | 1.5    | 27.31 | 26.32 | 26.15 | 26.20 | 25.89 | 25.81 | 25.83 | 26.34 |
> | 2.0    | 26.89 | 25.98 | 25.76 | 25.54 | 25.48 | 25.47 | 25.65 | 26.07 |
> | 2.5    | 26.83 | 26.04 | 25.83 | 25.58 | 25.51 | 25.48 | 25.54 | 25.98 |
> | 3.0    | 27.31 | 26.23 | 25.89 | 25.73 | 25.69 | 25.67 | 26.02 | 26.51 |
> | 3.5    | 27.53 | 26.82 | 26.32 | 26.23 | 25.92 | 25.96 | 26.15 | 26.78 |

---

> ### Author Response · Authors · 2025-11-23
> **Response to Reviewer J8YY**
>
> **[Effectiveness of subspace of geometry via visual analysis]** To provide more compelling evidences for supporting the idea of utilizing subspace geometry,  we visualize the CLIP’s visual features using UMAP, showing the low-dimensional, well-clustered nature for ID features, whereas dispersed, high dimensional nature for OOD-like background features (see Fig. 5 on page 8 of the revised manuscript). In addition, we also include qualitative visual analyses on a number of examples showing how **SubCoOp improves pseudo-OOD supervision (ID local patches vs OOD local patches separation) via its geometry-aware prompt learning**, which in turn accounts for its performance gains at test time **(see Fig. 6 on page 8 of the revised manuscript)**.
>
>
>
> **[Performance gain brought by SubCoOp]** Our experiments shows **enhancement brought by SubCoOp consistently across 4 different recent prompt learning OOD approaches** by noticeable margin. Our design is both computationally efficient and grounded in well-established principles of subspace geometry. Please see the table below:
>
>
> Table 4: Relative average FPR95 reduction brought by SubCoOp in each method.
>
> | Method | Average FPR95 Reduction |
> |----------|:----------:|
> | CoOp  | 6.8 \%  |
> | LoCoOp | 7.6 \%  |
> | SCT | 9.7 \%  |
> | OSPCoOp | 3.4 \%  |
>
>
>
> **[Orthogonality constraints]** Thank you for raising this point. We had investigated the effect of enforcing orthogonality constraints during development, but observed only marginal gains while incurring the extra cost of orthogonalizing the prompt vectors.

---

### Official Review · Reviewer_Ufmf · 2025-10-30

**Soundness:** 3
**Presentation:** 3
**Contribution:** 2
**Rating:** 4
**Confidence:** 3

**Summary:**

This paper presents a prompt optimization-based few-shot OOD detection method for pre-trained VLMs such as CLIP.

The proposed method builds upon the existing few-shot OOD detection framework called LoCoOp (Miyai et al., NeurIPS 2023), which extracts ID-relevant and ID-irrelevant regions from real ID images (foreground and background) and uses them as pseudo ID and OOD samples for prompt optimization; the goal of this work is to improve the discriminability between such regions.

To this end, the core idea is to introduce subspace regularization into the LoCoOp framework. Motivated by the observation that ID-relevant features tend to align with the subspace spanned by the learned prompt embeddings, the method optimizes the prompts so that pseudo ID regions lie within this subspace, while ID-irrelevant regions are projected onto its orthogonal complement.

Experimental results demonstrate that the proposed method achieves comparable or superior performance to LoCoOp-based baselines across multiple datasets.

**Strengths:**

**S1.** Although subspace-based approaches have been widely explored in the OOD detection literature, prior work within LoCoOp and its variants have not leveraged the prompt-spanned subspace and its orthogonal complement to separate ID-relevant and ID-irrelevant regions.

**S2.** The method is conceptually simple and well-motivated.

**S3.** The paper is clearly written and easy to follow.

**Weaknesses:**

**W1. Novlty**

Although the idea of leveraging subspace geometry for separating ID-relevant and ID-irrelevant regions is interesting, the contributions of this paper are somewhat limited in terms of novelty across idea, motivation, and method.

- *Idea*: While the introduction of subspace regularization is conceptually interesting, the use of subspace geometry for OOD detection has already been widely explored in prior work, e.g., [a–c].

- *Motivation*: Improving the quality and reliability of ID-relevant and ID-irrelevant regions in LoCoOp is a well-recognized objective. Indeed, similar goals have already been addressed by SCT (Yu et al., NeurIPS 2024) and OSPCoOp (Xu et al., CVPR 2025), both of which aim to address the same limitations of LoCoOp in selecting and diversifying ID-irrelevant regions.

- *Method*: The overall pipeline and components (e.g., local feature extraction and loss reweighting) remain almost identical to LoCoOp and SCT. Technically, the method can be regarded as adding a subspace regularization term on top of the SCT formulation.

Consequently, the proposed approach substantially depends on existing LoCoOp-based designs, and its contribution, though reasonable and well-motivated, appears relatively incremental.


**W2. Technical Soundness**

The proposed geometric regularization is reasonable and mathematically sound. However, the ID-irrelevant regions (proxy OOD regions) generated within the LoCoOp framework suffer from several inherent limitations:

- they are mostly confined to the background areas of ID images, resulting in limited diversity, especially in few-shot settings; and

- their selection is based on confidence scores, which can be affected by context bias; CLIP often assigns high confidence to background regions that frequently co-occur with foreground objects.

The proposed subspace regularization could potentially mitigate these issues by promoting better generalization, but this hypothesis is not thoroughly analyzed in the paper.


**W3. Empirical Significance**

**W3-1.** The empirical advantage of the proposed method is marginal. Compared with OSPCoOp, it improves AUROC by only +0.1% and reduces FPR95 by −0.9%; compared with SCT, the gains are +0.9% in AUROC and −2.7% in FPR95. These differences are small relative to the reported error bars and may easily vanish or reverse under different hyperparameter settings (Fig. 6–8).

**W3-2**. Several recent methods, such as [d] and [e], are not included in the comparison. Unless there is a clear justification for their exclusion, it would be desirable to include them for a fair and up-to-date evaluation.

**W3-3.** Given the small performance margin, the method is likely sensitive to hyperparameter choices. However, the sensitivity analysis is limited in scope: Fig. 6 explores a narrow range of values, and the interaction between $\lambda_1$ - $\lambda_3$ and $C$ is not examined. A more comprehensive and systematic evaluation of these parameters would strengthen the empirical evidence.

-----
[a] Guan et al., Revisit PCA-based technique for Out-of-Distribution Detection, ICCV 2023.

[b] Behpour et al., GradOrth: A Simple yet Efficient Out-of-Distribution Detection with Orthogonal Projection of Gradients, NeurIPS 2023.

[c] Zaeemzadeh et al., Out-of-Distribution Detection Using Union of 1-Dimensional Subspaces, CVPR 2021.

[d] Zeng et al., Local-Prompt: Extensible Local Prompts for Few-Shot Out-of-Distribution Detection, ICLR 2025.

[e] Zhang et al., LAPT: Label-driven Automated Prompt Tuning for OOD Detection with Vision-Language Models, ECCV 2024.

**Questions:**

**Q1. (Related to W1)**

If there are any factual misunderstandings in W1, please clarify them in the authors' rebuttal.

**Q2. (Related to W2)**

Could the authors provide objective evidence that the proposed subspace regularization mitigates the weaknesses of the LoCoOp framework? A more detailed investigation, such as qualitative visualization or correlation analysis between region selection and semantic context, would strengthen the technical claims.

**Q3. (Related to W3)**

Are there specific reasons why recent methods such as [d,e] were not included in the comparison? If not, could the authors provide results including these methods for a fair and up-to-date evaluation? Additionally, have the authors conducted broader or joint parameter sweeps to confirm the robustness of the reported results?

---

> ### Author Response · Authors · 2025-11-23
> **Response to Reviewer Ufmf**
>
> We would like to thank the reviewer for your reviewing efforts and insightful comments. Please see our detailed response as follows:
>
>
>
> **[Clarification on novelty and contributions]** We agree with the reviewer that subspace geometry has been previously explored in the context of OOD detection. However, our contribution lies in **systematically integrating subspace geometry into prompt-learning frameworks for OOD detection, which to the best of our knowledge has not been studied before**. We build on a LoCoOp-style framework primarily because it offers a simple yet effective way to obtain pseudo-OOD samples and is computationally more efficient than alternatives such as OSPCoOp ($\approx$ 4x more compute time) and Local-Prompt ($\approx$ 8x more compute time) that uses data augmentation strategies. More importantly, as we highlight in our paper, unlike other pseudo-OOD supervision methods such as LoCoOp, SCT, OSPCoOp, and Local-Prompt that rely solely on predicted probabilities, our approach also exploits the **geometry of ID and OOD embeddings**, enabling more effective separation of ID and OOD. This is validated in our experiments showing **enhancement brought by SubCoOp across 4 different state-of-the-art prompt learning OOD approaches (averaged over 3 random trials)**. See the following tables supporting our clarifications:
>
>
>
> Table 1: Training time for different prompt learning-based OOD approaches. We have used NVIDIA Tesla V100 GPU (32GB) for our experiments.
>
> | Method       | ID/OOD Data Aug. | Training time/epoch | Total training time |
> |:------------:|:----------------:|:-------------------:|:-------------------:|
> | CoOp         | $\times$         | 0.52 s              | 5 h 30 min          |
> | LoCoOp       | $\times$         | 0.66 s              | 6 h 10 min          |
> | SCT          | $\times$         | 0.66 s              | 3 h 10 min          |
> | SubCoOp      | $\times$         | 0.67 s              | 3 h 15 min          |
> | OSPCoOp       | $\checkmark$     | 2.6 s               | 11 h 38 min         |
> | Local-Prompt| $\checkmark$     | 6.1 s               | 24 h 5 min         |
>
>
>
>
>
> Table 2: Relative average FPR95 reduction brought by SubCoOp in each method
>
> | Method | Average FPR95 Reduction |
> |----------|:----------:|
> | CoOp  | 6.8 \%  |
> | LoCoOp | 7.6 \%  |
> | SCT | 9.7 \%  |
> | OSPCoOp | 3.4 \%  |
>
>
>
> **[Technical soundness: Qualitative visualization showing better generalization by SubCoOp]** Thank you for appreciating our idea and acknowledging the advantage of incorporating subspace regularization to enhance generalization. To further clarify our motivation behind the **idea of utilizing the low-dimensional subspace geometry**, we visualize the CLIP’s visual features using UMAP, showing the low-dimensional, well-clustered nature for ID features, whereas dispersed, high dimensional nature for OOD-like background features (see Fig. 5 on page 8 of the revised manuscript). In addition, we also include qualitative visual analyses on a number of examples showing how **SubCoOp improves pseudo-OOD supervision (ID local patches vs OOD local patches separation) via its geometry-aware prompt learning**, which in turn accounts for its performance gains at test time **(see Fig. 6 on page 8 of the revised manuscript)**.

---

> ### Author Response · Authors · 2025-11-23
> **Response to Reviewer Ufmf**
>
> **[Additional baselines]** As suggested by the reviewer, we now compare with additional baselines such as Local-Prompt that uses data augmentation strategies. SubCoOP outperforms Local-Prompt in 4-shot settings in both ImageNet-1K and ImageNet-100 datasets.
>
> Table 3: Performance analysis of ImageNet-1k dataset with our SubCoOp method with 4-shot setting.
> | **Method**              | **FPR95** | **AUROC** |
> |------------------------|---------|-----------|
> | LoCoOp                 | 30.75   | 92.70     |
> | SCT                    | 30.22   | 92.82     |
> | Local-Prompt+LoCoOp    | 27.85   | 97.47     |
> | SubCoOp           | **26.96** | **97.61** |
>
>
> Table 4: Performance analysis of ImageNet-100 dataset with our SubCoOp method with 4-shot setting.
> | **Method**              | **FPR95** | **AUROC** |
> |------------------------|---------|-----------|
> | LoCoOp                 | 16.56   | 96.68     |
> | SCT                    | 17.32   | 96.56     |
> | Local-Prompt+LoCoOp    | 16.21   | 96.93     |
> | SubCoOp           | **14.85** | **97.27** |
>
>
> In 16-shot setting, SubCoOp remains competitive, yet Local-Prompt combined with LoCoOp has a slight advantage at the cost of approximately 8–9× more computation.
>
>  Table 5: OOD performance and Training time with 16-shot setting for different prompt learning-based OOD approaches. We have used NVIDIA Tesla V100 GPU (32GB) for our experiments
>
> | Method | FPR95 | AUROC | Total Training time
> |----------|----------|----------|----------|
> | LoCoOp | 30.02  | 93.14  | 6h 10 min |
> | SubCoOp  | 25.47  | 94.07  | 3h 15 min |
> | Local-Prompt| 28.27  | 93.58  | 24h 5 min |
> | Local-Prompt+LoCoOp| 24.52  | 94.48  | 26h 15 min |
>
>
> The methods such as LAPT rely on outlier-label exposure and require external label sources (e.g., large text corpora) for negative label mining. On the other hand, our method focuses on pseudo-OOD prompt-learning setting without external resources.  We have clarified this distinction in the revised manuscript in the related works section.

---

> ### Author Response · Authors · 2025-11-25
> **Response to Reviewer Ufmf**
>
> **[Hyperparameter sensitivity and more ablation studies]** Regarding the sensitivity of the approach, we would like to highlight that all our results (including hyperparameter study and baseline comparisons) are averaged over 3 random trials, and the standard deviation is also reported. In particular, in Fig 8-10, the performance of our approach is quite stable for a large range of hyperparameter values. To further support this, as suggested by the reviewer, we are expanding our ablation study over further larger ranges and also adding analysis of interactions between our subspace regularizations $\lambda_1$ and $\lambda_2$.
>
>
> Table 6: Average detection performance of SubCoOp method in FPR with $\lambda_1$ and $\lambda_2$ using ImageNet-1k as ID dataset.
>
> | λ1 / λ2 | 0     | 1     | 2     | 3     | 4     | 5     | 6     | 7     |
> |--------|-------|-------|-------|-------|-------|-------|-------|-------|
> | 0.0    | 28.23 | 28.16 | 27.76 | 27.53 | 27.19 | 27.33 | 27.51 | 27.32 |
> | 0.5    | 27.96 | 27.47 | 27.52 | 27.18 | 26.87 | 26.84 | 27.12 | 27.05 |
> | 1.0    | 27.62 | 26.71 | 26.44 | 26.32 | 26.26 | 26.54 | 26.71 | 26.86 |
> | 1.5    | 27.31 | 26.32 | 26.15 | 26.20 | 25.89 | 25.81 | 25.83 | 26.34 |
> | 2.0    | 26.89 | 25.98 | 25.76 | 25.54 | 25.48 | 25.47 | 25.65 | 26.07 |
> | 2.5    | 26.83 | 26.04 | 25.83 | 25.58 | 25.51 | 25.48 | 25.54 | 25.98 |
> | 3.0    | 27.31 | 26.23 | 25.89 | 25.73 | 25.69 | 25.67 | 26.02 | 26.51 |
> | 3.5    | 27.53 | 26.82 | 26.32 | 26.23 | 25.92 | 25.96 | 26.15 | 26.78 |
>
>
>
>
>  Notably, SubCoOp’s improvements are stable across different hyperparameter configurations, suggesting that the method is robust and not overly sensitive to hyperparameters. Here, when both $\lambda_1$ and $\lambda_2$ is 0, it is considered as SCT in Table 4.
>
>
>
>
> In the revised manuscript, we substantially extend our sensitivity analysis of our SubCoOp method by exploring a wider range of values for $\lambda_1$, $\lambda_2$, $\lambda_3 \$ and $C\$, and their interactions as presented in the newly added **Figure 13 in page 17**.

---

### Comment · Area_Chair_e1qT · 2025-11-20
**To AI Review**

Dear authors,

I would like to share an important reminder regarding the review process. Recently, we have noticed that some reviewers may be using AI tools to help generate their reviews. This can lead to low-quality or inaccurate feedback, which is unfair to authors who deserve careful and thoughtful evaluations.

To help maintain fairness, I kindly ask for your assistance: If you believe a review you received was partly or fully generated by AI, and you have some evidence (for example: unusual writing style, clear factual mistakes, AI-detector results, repeated generic sentences, etc.), please feel free to contact me directly.

I will review any evidence you provide and, if appropriate, adjust the weight of the reviewer’s evaluation so that it does not negatively affect your submission. Thank you for helping us keep the review process fair and responsible. Your understanding and cooperation are greatly appreciated.

Best regards,

AC

---

### Comment · Area_Chair_e1qT · 2025-11-27

Dear Reviewers and Authors,

As we are approaching the rebuttal deadline, I would like to share a gentle reminder with everyone.

For authors:
If you have not yet submitted your rebuttal, please make sure to do so as soon as possible. Submitting very close to the deadline may reduce the chance for reviewers to read and respond in time, which could affect the discussion phase.

For reviewers:
If a rebuttal has already been submitted for your assigned paper, I encourage you to take a moment to read it and, where appropriate, provide a brief response or update your evaluation. Of course, this is not meant to pressure anyone into changing scores, it is simply to ensure that all reviews remain well-informed before final decisions.

Thank you all for your time and effort in keeping the review process smooth and constructive.

Warm regards,
AC

---

### Author Response · Authors · 2025-11-27
**Summary of our response**

Dear Reviewers,

Thank you again for your reviewing efforts, careful evaluation, and constructive feedback. We hope our responses addressed your concerns and helped clarify the strengths and contributions of our work. Please let us know if you have more questions and we are happy to address further.

 Meanwhile, we provide a brief summary of our overall response for your reference. We list the major questions raised in your reviews and the corresponding summary of our response.

1. ***Clarification of novelty and contributions***
   - We further clarified the key novelty and main contributions of our work and also clarified how does **the existing prompt learning based methods do not accommodate the distinctive geometry of the ID and OOD samples which is central to our method**.
   - We added visual explanations using UMAP figures and ID/OOD local region extraction examples of our method (see Figs. 5 and 6 in the revised manuscript).
   - We highlighted the **consistent enhancement in OOD detection performance brought by our approach SubCoOp across different state-of-the-art prompt learning OOD approaches and different few-shot settings.**

2. ***Expanded hyperparameter study***
   - We expanded our **hyperparameter study to cover a wider range of subspace regularization coefficients** $\lambda_1\$ and $\lambda_2\$ and numbers of prompt vectors \(M\), demonstrating the stability of our approach.
   - We added **joint hyperparameter sweeps** (see Fig 13 in supplementary material of the revised version).

3. ***Additional baselines and datasets***
   - We presented evaluation on an **additional prompt-learning approach LocalPrompt, further demonstrating the computational efficiency of SubCoOp and its effectiveness even without using any external knowledge for OOD supervision**.
   - We added **experiments on the OpenImage-O dataset, and we also included results on ImageNet-10 as ID and ImageNet-100 as OOD**.
   - We expanded the discussion of additional baselines and clarified their distinctions from our method.

4. ***Manuscript revision and organization***
   - We updated the **contributions section to better reflect the novelty of our work** and its differences from existing methods.
   - We **reorganized the experimental section** to improve readability.
   - We **added more figures illustrating the effectiveness of our method (fig. 5 and 6)**, as well as additional results in figures/tables **(Fig 8, Fig. 13 and Table 3) reporting the hyperparameter studies** and evaluation results.


Sincerely,
Authors

---

### Meta-Review · Area_Chair_94iE · 2026-01-05

**Summary:**

The paper initially received mostly negative scores: 6, 4, 4, 2. The main concerns include: (1) limited contribution and novelty; (2) marginal improvement; (3) insufficient comparisons and discussions from related methods; and (4) missing hyper-parameter analysis and evaluations on more datasets.

The authors have provided a detailed rebuttal to respond to the reviewers’ concerns. The AC has carefully read the reviews and the rebuttal, and finds that the authors have addressed concerns (3) and (4), along with other minor points.

However, the critical concerns, i.e., (1) and (2), are not well resolved. First, most components of the proposed method are similar to LoCoOp and SCT, and the main goals of this work have been solved/considered in previous methods. (2) in several cases, the improvement of the proposed method is marginal.

Given these considerations, the AC believes the critical concerns were not adequately addressed in the rebuttal, and that most reviewers are unlikely to change their original scores. The AC thus regretfully recommends rejection and hopes the authors can address the remaining concerns in a future submission.

**Reviewer Concerns:**

Solved Concerns: insufficient comparisons and discussions from related methods; missing hyper-parameter analysis and evaluations on more datasets; other minor concerns.

Unsolved Concerns: (1) limited contribution and novelty; (2) marginal improvement

**Reviewer Scores:**

Reviewer Ufmf  and Reviewer J8YY will not change their scores as the concerns of novelty and improvement were not solved.

Reviewer KQBt may change his/her score from 2 to 4, as some of the concerns were solved. However, the concern of novelty is not solved.

Reviewer yiNu will keep his/her score of 6 as the corresponding concerns were solved.

---

### Decision · Program_Chairs · 2026-01-26

Reject